# TOPOLOGY OVER BIOLOGY: NETWORK REPRESENTATION IMPROVES MULTI-OMICS MODELS WITHOUT NEED FOR PRIOR KNOWLEDGE

## ABSTRACT

Cancer is a heterogeneous and complex disease with substantial variation in patient outcomes. Multi-omics data (including mRNA expression, DNA methylation and micro-RNA expression) capture transcriptional and post-transcriptional regulation of gene expression within the tumor microenvironment, with the potential to reveal mechanisms responsible for different patient outcomes. However, multi-omics data are complex and high dimensional, and extracting meaningful features through machine learning is a challenging task. Current SOTA techniques involve GNNs based on correlation networks built using omics data, and more recent models introduce improvements by augmenting these correlation networks with known biological interactions and pathways. However, this approach relies on the experimental characterization of biological interactions, which requires significant resources. In this work, we take a different approach by enhancing the representation of the correlation networks using topological tools: the Mapper algorithm for pooling nodes, and topological deep learning to represent higher order interactions. Our novel biology-agnostic models M-SAN and M-HGAT outperform both the naive correlation network approach, and models augmented with prior knowledge, in survival prediction across six cancer types (breast cancer, colon cancer, kidney cancer, melanoma, lung cancer and ovarian cancer) with sample sizes between 149 and 333. Additionally, by examining the most important feature interactions within our models, we find that they have learned gene interactions corresponding to biological processes relevant to cancer proliferation and metastasis.

## 1 INTRODUCTION

Advances in biological sequencing have led to increasing interest in machine learning for precision medicine, which aims to identify more effective treatments based on patient sequencing data. Multi-omics combines data from different "omes" (e.g. genome, epigenome, transcriptome) to provide a complete molecular characterization of disease, with the potential to uncover new biomarkers and disease pathways. However, multi-omics data are complex and high-dimensional, and learning meaningful representations with machine learning models is a challenging task.

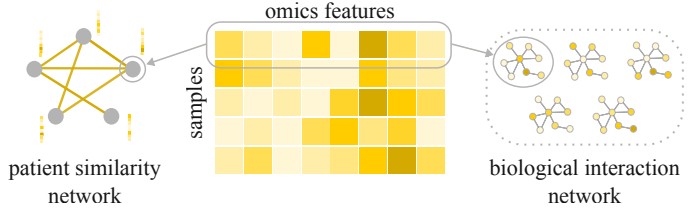

Figure 1: Construction of a patient similarity or biological interaction network using omics data.

Current SOTA models in supervised learning on multi-omics data utilize graph neural networks (GNNs). These are based on correlation networks, and fall into the two categories shown in Figure 1. For each omic type, data can be represented in the form $X \in \mathbb{R}^{N \times K}$, where $N$ is the number of patients and $K$ is the number of omics features. A patient similarity network (Wang et al., 2021; Tanvir et al., 2024; Alharbi et al., 2025) can be constructed by representing each patient as a node, and adding edges based on pairwise row correlations (using a cut-off value) between feature vectors $X_{i,:}, X_{j,:} \in \mathbb{R}^K$ for each pair of patients, $i$ and $j$. The omics features for each patient become node features, and patient classification is a node classification task. Alternatively, a biological interaction network Xing et al. (2021); Zhu et al. (2023); Hussein et al. (2024) can be constructed by representing each feature as a node, and adding edges based on column-wise correlations for features across all patients $X_{:,\alpha}, X_{:,\beta} \in \mathbb{R}^N$ for each pair of features, $\alpha$ and $\beta$. Each patient is represented by a graph (with the same connectivity), and node features are the patient's omics features. Patient classification in this approach is therefore a graph classification task.

More recent models often augment or replace these networks using prior knowledge in the form of protein-protein interaction networks, or gene pathway information (Tan et al., 2025; Xiao et al., 2023; Alharbi et al., 2025). However, this does not always boost performance (Alharbi et al., 2025), and use of general networks may not be appropriate as protein-protein interactions are known to be tissue and disease dependent (Li et al., 2024). Moreover, this approach cannot yield models which advance biological understanding by uncovering previously unknown interactions.

We instead take a prior-knowledge agnostic approach to improving GNN-based multi-omics models, by targeting two main issues that arise in correlation networks constructed from omics data. The first is that despite careful selection of the correlation cut-off used to construct the edges, regions of the graph may still be very densely connected. This can lead to oversmoothing as nodes share many common neighbors (Hossain et al., 2024; Wu et al., 2023), and is particularly problematic in biological interaction networks, which can have tens of thousands of nodes. The second issue is that graphs can only capture pairwise interactions, whereas higher order interactions may be important, particularly in biological interaction networks, where groups of several genes frequently act together.

To address these issues, we use two tools from the fields of topological data analysis and topological deep learning to refine both patient similarity and biological interaction GNN models. We evaluate our new models on their performance in predicting cancer survival, focusing on cancer as a heterogeneous disease, with large variations between individuals in quality of life and survival outcomes. We focus on survival rather than the commonly used task of subtype prediction, as survival is a holistic measurement that incorporates factors such as cancer stage, treatment response, and overall patient health. Accurate survival prediction therefore requires a more comprehensive representation of disease biology compared to subtype prediction, which may have explicit signatures that can be directly inferred from the omics features (Prat et al., 2012). We show that especially in the case of biological interaction networks, our tools produce consistent increases in performance in predicting survival outcomes across all six cancer types, and as our methods do not rely on biological knowledge, they are expected to generalize to similar network-based problems in other domains.

In this work, we make the following **main contributions**:

- Introduce the Mapper algorithm and topological deep learning (including a novel simplicial attention network architecture that allows message passing between cells of adjacent ranks) to improve performance of both patient similarity and biological interaction GNN models.
- Show that our improved models, M-SAN and M-HGAT, perform better than current SOTA models for multi-omics classification, and outperform cancer survival models leveraging protein-protein interaction and gene pathway information.
- Demonstrate that our models assign high feature importance to gene interactions that correspond to biological processes associated with cancer proliferation and metastasis, often specific to the cancer type, suggesting successful representation of cancer biology despite having no access to biological prior knowledge.

## 2 BACKGROUND

**The Mapper algorithm** is a tool from the field of topological data analysis which emphasizes the shape of data (Singh et al., 2007). While it is typically used as a tool for visualization of high dimen-

sional data, it also performs dimensionality reduction by constructing a Mapper graph comprised of nodes representing clusters of data computed in the high dimensional space. We use Mapper in our work to convert omics data into a coarse-grained patient similarity network or biological interaction network which are input into our model architectures.

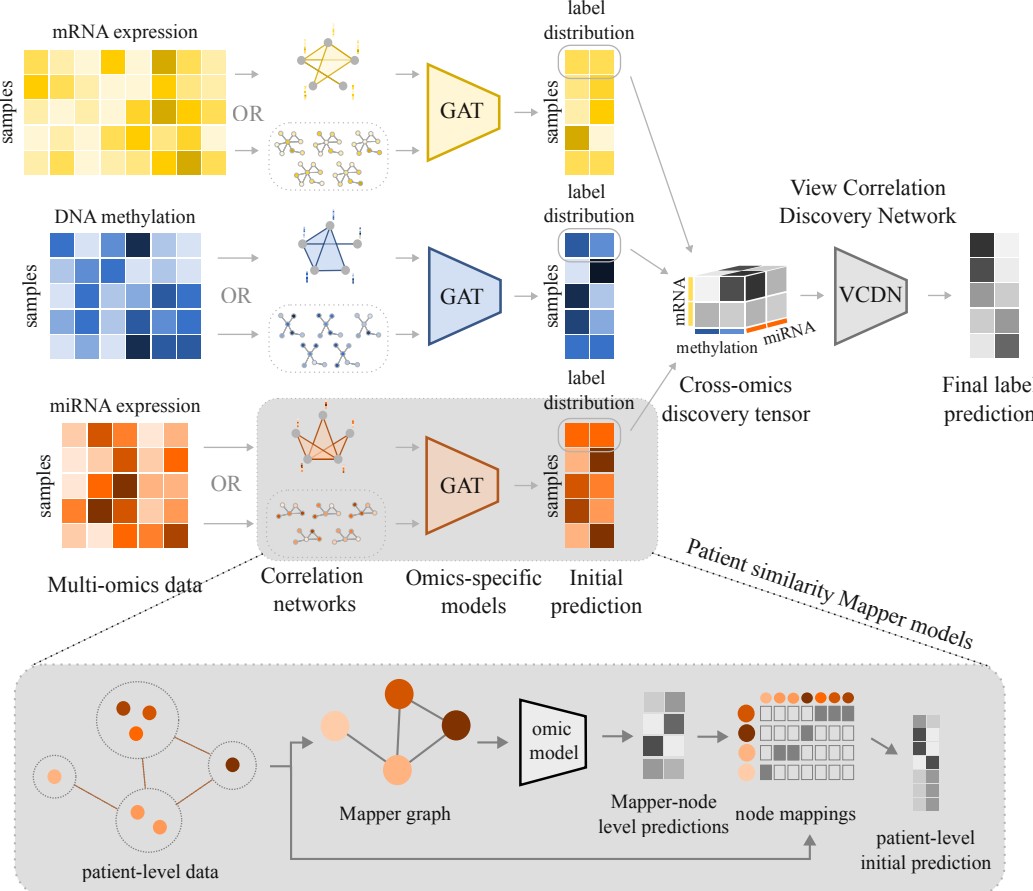

Figure 2: Main: Benchmark GNN architecture using patient similarity networks (top network in the correlation networks column), or biological interaction networks (bottom networks in the correlation networks column). Figure adapted from Wang et al. (2021) in accordance with the terms of Creative Commons Attribution 4.0 International License. Inset: The Mapper algorithm is integrated into the patient similarity models by replacing the gray portion of the benchmark GNN with the pipeline shown in the inset. Patients are grouped into nodes on a Mapper graph where each node can contain more than one patient, in which case the node features are the averaged features of the patients within the node. The node mappings that are used to produce the Mapper graph are applied to the output of the omics-specific model to obtain patient-level label predictions.

**Topological deep learning** (TDL) is a collection of deep learning models defined on topological domains including simplicial complexes, cellular complexes, combinatorial complexes and hypergraphs, which contain higher-order interactions beyond the pairwise interactions modeled by graphs (Hajij et al., 2022). They can be viewed as an extension to GNNs as they can model interactions between groups of more than two nodes. In this work, we focus on the **simplicial attention network** (SAN) and **hypergraph attention network** (HyperGAT) as attention-based message passing models on two main classes of topological structures (Papillon et al., 2023). The SAN is defined on simplicial complexes comprising hierarchical part-whole relations constructed through multi-rank objects called cells. Rank 0 cells are nodes, rank 1 cells are edges, rank 2 cells are triangles, and rank 3 cells are tetrahedrons. Hypergraphs comprise non-hierarchical set-type relations, where a hyperedge can exist between any number of nodes, and all have equal importance.

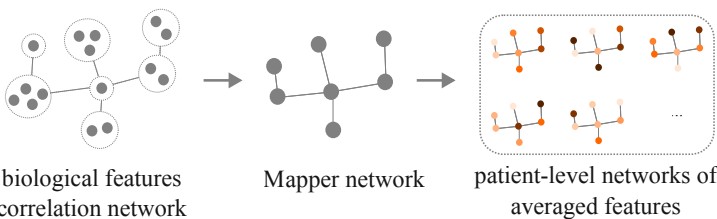

biological features
correlation network

Mapper network

patient-level networks of
averaged features

Figure 3: The Mapper biological interaction network. Biological features are grouped into nodes in a Mapper graph, and the features within a node are averaged for each patient.

## 3 METHODS

### 3.1 BENCHMARK GNN MODELS

We base our benchmarks on the adapted version of MOGONET (Wang et al., 2021) which was the top performer in a comparison of deep learning models for multi-omics (Leng et al., 2022). We replace the graph convolutional networks (GCNs) used in MOGONET with graph attention networks (GATs), as this modification has been shown to improve performance (Leng et al., 2022; Hussein et al., 2024; Alharbi et al., 2025). As illustrated in Figure 2 (main panel), a patient similarity network or a set of biological interaction networks is constructed from each omics dataset. These are passed through omics-specific GATs which outputs class probabilities for each sample via the node-level outputs for the patient similarity network, or through a global mean pooling layer applied to the node-level outputs in the case of the biological interaction networks. These are combined along a new dimension for each omic type, and input into a View Correlation Discovery Network (VCDN) (Wang et al., 2019), which outputs the final class probabilities.

### 3.2 MAPPER AND TDL MODELS

Figures 2 inset and 3 show the changes made in the Mapper models, and Appendix A.1 contains further details. For the patient similarity network, Mapper may combine several patients into a single node on the Mapper graph. Therefore, to obtain the patient-level initial prediction from each of the omics-specific models, we apply a precomputed mapping from Mapper nodes to patient samples to the output of the omics-specific models. For patients in the test set which were not included in the construction of the Mapper graph, we use a predictive Mapper algorithm (Lee & Jung, 2023). In brief, this algorithm registers the cluster boundaries of the preimage when the Mapper graph is computed, and uses these to determine the appropriate node assignment of unseen data points. New patients are thus assigned either to a pre-existing Mapper node, or classifies as not belonging to any existing node, in which case they are appended as a new node to the Mapper graph, with connecting edges constructed based on pair-wise distance correlations (using the same threshold as when constructing the Mapper network used for training, see Appendix A.1) between the new patient's features and averaged features of the existing Mapper nodes. The initial predictions from the omics-specific models are combined into a cross-omics discovery vector and fed into the VCDN for the final label predictions exactly as in the benchmark GNNs. For the biological interaction network, no changes to the architecture are required to incorporate the Mapper algorithm. Mapper simply produces a coarse-grained version of the correlation network, which can be input directly into the omics-specific models. In the TDL models, the omics-specific GAT in the benchmark GNNs is replaced with either a SAN or HyperGAT module.

**Mapper SAN**: While several formulations of SANs exist (Goh et al., 2022; Giusti et al., 2022), they do not allow for message passing between cells of different ranks. We instead build our own module based on the simplicial complex convolutional network from the TopoModelX package (Yang et al., 2022; Hajij et al., 2024), modifying the message passing scheme to incorporate attention. We describe our novel message passing scheme in this section, and Appendix A.2 contains additional module details.

For each omics Mapper network, we have feature matrices $X^0, X^1, \ldots, X^{\text{max rank}}$, where the superscript denotes the rank of the cells. $X^0$ represents the node features, and the features for the higher ranks are initialized as the gradient of the feature matrix one rank lower. The gradient is given by the product $\mathbf{B}_{r+1}^T X_r$, where $X_r$ is the feature matrix of rank $r$, and $\mathbf{B}_{r+1}$ is the incidence matrix of rank $r + 1$ having a number of rows equal to the number of simplices of rank $r + 1$, and number of columns equal to the number of simplices of rank $r$. The incidence matrix establishes which rank $r + 1$ simplices are in the neighborhood of which rank $r$ simplices, with 0 entries denoting no incidence and 1 entries denoting incidence (see Battiloro et al. (2024) for further details).

The neighborhood of a cell contains cells of the same rank, cells one rank lower and cells one rank higher. $x$ and $y$ are neighbors if $x$ is on the boundary ($r_1 < r_2$) or co-boundary of $y$ ($r_1 > r_2$), or if $x$ and $y$ are upper or lower adjacent ($r_1 = r_2$). To illustrate, if $x$ is an edge, $y$ would be in its boundary adjacent neighborhood if $y$ were a node connected to $x$; its co-boundary adjacent neighborhood if $y$ were a face that contained the edge $x$; upper adjacent if $y$ were an edge that shared a face with $x$; and lower adjacent if $y$ were an edge that was connected to the same node as $x$ (Papillon et al., 2023). Additionally, self-loops are added in the adjacency matrices (analogous to GNNs) to allow cells to pass messages to themselves.

Attention coefficients for message passing between neighboring cells $x$ (of rank $r_1$) and $y$ (of rank $r_2$) are calculated as:

$$\alpha_{xy} = \text{softmax}(W_a[v_x||v_y]) \tag{1}$$

where $W_a \in \mathbb{R}^{2K}$ (where $K$ is the number of features) are the learned attention weights, and $||$ denotes concatenation, and $v_x$ and $v_y$ are the features of cells $x$ and $y$.

The contribution to cell $y$ from neighboring cells of rank $r_1$ is given by:

$$v_y^{r_1'} = \text{RELU}\left(\sum_{x \in \mathcal{N}(r_1, r_2)} \alpha_{yx} W^{r_1} v_x\right) \tag{2}$$

where $W^{r_1}$ is a matrix of learned weights and $\mathcal{N}(r_1, r_2)$ is the rank-appropriate neighborhood matrix. The final features for cell $y$ in the next layer is obtained by summing the contributions from the ranks $r_2 - 1$, $r_2$ and $r_2 + 1$, and applying the RELU function again.

$$v_y' = \text{RELU}\left(v_y^{r_2-1'} + v_y^{r_2'} + v_y^{r_2+1'}\right) \tag{3}$$

**Mapper HyperGAT**: We use the HyperGAT module implemented in TopoModelX (Ding et al., 2020; Hajij et al., 2024). The mathematical formulation of this module can be found in Appendix A.3.

Table 1: Sample size and number of features by cancer type

| Cancer | Number of samples | | | Number of features | | |
|---|---|---|---|---|---|---|
| | Total | < median | ≥ median | mRNA | DNA Meth | miRNA |
| Breast | 277 | 61 | 216 | 18790 | 5000 | 929 |
| Colon | 170 | 22 | 148 | 18790 | 5000 | 613 |
| Kidney | 149 | 34 | 115 | 18790 | 5000 | 929 |
| Melanoma | 333 | 105 | 228 | 18790 | 5000 | 929 |
| Lung | 268 | 72 | 196 | 18790 | 5000 | 929 |
| Ovarian | 213 | 86 | 127 | 18790 | 5000 | 613 |

### 3.3 DATA AND PRE-PROCESSING

We used publicly available cancer datasets extracted from The Cancer Genome Atlas Program (TGCA).[1] This dataset contains 3 complementary omics datasets of mRNA expression data, DNA methylation data, and micro-RNA expression data from the TGCA accessed through the Broad GDAC Firehose with survival data obtained from Xena, and performs early pre-processing as outlined in Rappoport & Shamir (2018). We performed additional pre-processing by removing samples which do not have all three omics types and features with low variance across all cancer types which share common features. Survival times were dichotomized to avoid censorship bias by binning patients into one class that lived for less than the mean survival time for each cancer type, and one class that lived for at least the median survival time Leung et al. (1997). Median survival times were calculated using patients who were deceased, and patients who were not deceased, but had not yet lived for at least the median survival time were not included as samples. Table 5 in Appendix A.4 shows the median survival times computed for each cancer type.

Samples were split using stratified sampling into a 70-30 training/validation set (which was further split through an 80-20 split into training and validation sets) and test set. The features of the training/validation set were scaled to have a mean of zero and standard deviation of 1, and test set features were scaled using the means and standard deviations of the training/validation set. The number of features and sample size for each cancer type is shown in Table 1.

### 3.4 EXPERIMENTS

For each of the six cancer survival tasks, we evaluated the performance of eight models: the benchmark GNN (B-GNN), the Mapper GNN (M-GNN), the Mapper SAN (M-SAN) and the Mapper HyperGAT (M-HGAT), using both the patient similarity network and biological interaction networks. Training used the Adam optimizer and weighted cross entropy loss to account for class imbalance in the data. Further details about training resources and hardware are detailed in Appendix A.5 together with the hyperparameter tuning procedures and optimal hyperparameter values for each model in each cancer type. For performance measures, we use the ROC-AUC (also commonly known as the $c$-statistic or $c$-index in survival analysis), and the F1 score. To prevent inflation of the F1 score due to dataset imbalance, we label the minority class as positive. Performance measures are averaged over five independent runs, and the error is reported as one standard error of the mean.

In addition to the two benchmark GNNs, we also include a baseline of a simple MLP classifier, with one single layer for dimensionality reduction of each of the omics types, followed by concatenation of the omics features, which are passed into a final single-layer MLP that outputs the class labels.

## 4 RESULTS

### 4.1 CANCER SURVIVAL PREDICTION

The performance of each model on each of the cancer types is shown in Table 2, and Table 3 is a summary table of model performance averaged across all the cancer types. Significance testing of c-indexes (ROC-AUC values) using a t-test with Bonferroni-adjusted p-values shows that the best model performed significantly better (p-value $< 0.05$) than both benchmark GNNs (indicated by ** in Table 2) for breast cancer, and kidney cancer. The best model for lung cancer, colon cancer and ovarian cancer (and several other Mapper/TDL models for breast cancer, lung cancer, kidney cancer and ovarian cancer) performed significantly better than their corresponding benchmark GNN (i.e., when comparing only among the patient similarity models or the biological interaction models, indicated by * in Table 2). For most cancer types, one of the Mapper models, particularly the TDL models of Mapper SAN or Mapper HyperGAT, is the best model, and this is reflected in the summary table, where Mapper SAN is the top patient similarity model, and Mapper HyperGAT is the top biological interaction model and top overall.

Previous multi-omics GNNs were mainly used to predict cancer subtypes, however GGNN by Zhu et al. (2023), which incorporated protein-protein interactions and gene pathways data, was used to

---

[1] http://acgt.cs.tau.ac.il/multi_omic_benchmark/download.html

predict cancer survival using TCGA data. The $c$-index values they report have been included in Table 2, and the averaged value reported in Table 3. While the utilization of prior knowledge enables GGNN to often outperform the benchmark models, our topological models consistently outperform GGNN, often by a large margin. This suggests that topological tools can be more successful than biological priors at arriving at a network structure that supports the learning of useful representations.

Additionally, the performance of non-GNN models including CustOmics Benkirane et al. (2023) – a variational autoencoder-based multi-omics model, and BulkRNABert Gélard et al. (2025) – an LLM-based single-omics model using mRNA features are included in Table 2 as available (for breast cancer and lung cancer), and can be seen to yield worse performance than the top Mapper-TDL models for these cancers. All models perform better than the simple MLP classifier.

## 4.2 PATIENT SIMILARITY MODEL PERFORMANCE

When the Mapper algorithm combines several patients into a single node on the Mapper graph, their features are averaged to obtain the node features, resulting in a loss of individual patient data. Surprisingly despite this, the Mapper GNN and TDL models perform better than the benchmark GNN on average. In this section, we investigate why this is the case, and the conditions under which our topological tools are the most effective.

We split the cancers into two groups based on whether the Mapper GNN outperforms the benchmark GNN to compare and contrast the effectiveness of the Mapper algorithm within and between these groups. Table 2 shows that the Mapper GNN performs worse than the benchmark GNN on at least one metric for breast cancer, colon cancer, and melanoma (which we refer to as the benchmark group), whereas for kidney, lung and ovarian cancer (the Mapper group), the Mapper GNN performs better than the benchmark GNN on both metrics. To quantify the effectiveness of the Mapper algorithm for a given cancer type, we apply GNNExplainer (Ying et al., 2019) to each node on the benchmark patient-level graph, to identify the patients who had nonzero contributions to obtaining the correct label prediction for each patient. We then compared these results to the Mapper graph, to see which of these patients had been combined into the same Mapper node. This was carried out for each of the three omics-specific GNNs, and Table 9 in Appendix A.6 shows the fraction of patients that are combined into Mapper graph nodes with samples that are important for explaining their label. To aggregate across the three omics types, we compute the mean, minimum, median and maximum, and show that all these aggregated values are higher on average for the Mapper group than the benchmark group. Therefore, the Mapper algorithm is able to group patients who are important in obtaining the correct label for each other, and in half of the cancer types, the benefits of this unsupervised node pooling outweighs the disadvantages of losing individual patient level features.

Similarly, we observe that TDL results vary across the different cancer types. For breast, colon and ovarian cancer (the TDL group), both the Mapper TDL models (M-SAN and M-HGAT) perform better than the Mapper GNN, whereas for kidney cancer, melanoma and lung cancer (the GNN group), at least one of the TDL models performs worse than the Mapper GNN. We apply GNNExplainer this time to the nodes of the Mapper GNN, to obtain the Mapper nodes important for predicting the correct label, and compare these to the Mapper nodes that are grouped together by higher order simplices/hyperedges in the TDL models (these are the same as the TDL networks are constructed in the same way). Table 10 in Appendix A.6 shows the fraction of higher order simplices/hyperedges that contain nodes important for explaining their labels. Again, this is done for each of the three omic types, and aggregated, and we find that all aggregations are on average higher for the TDL group than for the GNN group, showing that the effectiveness of TDL depends on combining nodes useful for generating the correct label for each other into higher order simplices/hyperedges.

## 4.3 BIOLOGICAL INTERACTION MODELS AND FEATURE IDENTIFICATION

In the biological interaction network, the Mapper algorithm combines features instead of samples, and helps to reduce the size of the network and sparsify densely connected regions. This is effective in improving performance, as the Mapper GNN outperforms the benchmark GNN across all six cancer types. In most cases, the Mapper TDL models further improve performance. Interestingly, the Mapper HyperGAT is the best model on average, as biological interactions follow natural set-type relations, where a number of equally important biomolecules act together to produce an effect

Table 2: Performance of patient similarity and biological interaction benchmark, Mapper models, and MLP baseline on predicting cancer survival. The results of GGNN (Zhu et al., 2023) which utilizes prior-knowledge networks, CustOmics (Benkirane et al., 2023) which utilizes variational autoencoders, and BulkRNABert (Gélard et al., 2025) which utilizes an LLM are also reported where available. Two asterisks on a model indicates that it had a significantly higher c-index/ROC-AUC (Bonferroni-adjusted t-test p-value $< 0.05$) than both the patient similarity and biological interaction benchmark GNNs, and a single asterisk on the other models indicates that it had a significantly higher c-index/ROC-AUC than the benchmark GNN of the same category (patient similarity or biological interaction).

| | | Model | ROC-AUC | F1 | | | Model | ROC-AUC | F1 |
|---|---|---|---|---|---|---|---|---|---|
| **Breast Cancer** | | MLP | 0.514 ± 0.025 | 0.322 ± 0.045 | **Lung Cancer** | | MLP | 0.556 ± 0.015 | 0.464 ± 0.019 |
| | | GGNN | 0.661 | - | | | GGNN | 0.563 | - |
| | | CustOmics | 0.642 ± 0.018 | - | | | CustOmics | 0.625 ± 0.037 | - |
| | | BulkRNABert | 0.604 ± 0.032 | - | | | BulkRNABert | 0.648 ± 0.057 | - |
| | Patient Sim. | B-GNN | 0.609 ± 0.008 | 0.411 ± 0.014 | | Patient Sim. | B-GNN | 0.602 ± 0.009 | 0.403 ± 0.024 |
| | | M-GNN | 0.556 ± 0.011 | 0.351 ± 0.017 | | | M-GNN* | 0.664 ± 0.007 | 0.555 ± 0.009 |
| | | M-SAN | 0.597 ± 0.005 | 0.391 ± 0.011 | | | **M-SAN*** | **0.685 ± 0.015** | **0.585 ± 0.018** |
| | | M-HGAT | 0.607 ± 0.012 | 0.417 ± 0.021 | | | M-HGAT | 0.623 ± 0.010 | 0.483 ± 0.008 |
| | Biological Int. | B-GNN | 0.529 ± 0.008 | 0.278 ± 0.038 | | Biological Int. | B-GNN | 0.635 ± 0.017 | 0.534 ± 0.008 |
| | | M-GNN | 0.532 ± 0.018 | 0.241 ± 0.052 | | | M-GNN | 0.647 ± 0.003 | 0.541 ± 0.003 |
| | | M-SAN* | 0.594 ± 0.007 | 0.434 ± 0.011 | | | M-SAN | 0.546 ± 0.007 | 0.272 ± 0.048 |
| | | **M-HGAT**** | **0.673 ± 0.003** | **0.500 ± 0.003** | | | M-HGAT | 0.679 ± 0.007 | 0.569 ± 0.013 |
| **Kidney Cancer** | | MLP | 0.452 ± 0.038 | 0.181 ± 0.056 | **Melanoma** | | MLP | 0.501 ± 0.010 | 0.354 ± 0.031 |
| | | GGNN | 0.690 | - | | | GGNN | 0.604 | - |
| | Patient Sim. | B-GNN | 0.670 ± 0.017 | 0.446 ± 0.019 | | Patient Sim. | **B-GNN** | **0.634 ± 0.007** | 0.470 ± 0.010 |
| | | M-GNN* | 0.766 ± 0.015 | 0.561 ± 0.019 | | | **M-GNN** | 0.623 ± 0.006 | **0.477 ± 0.006** |
| | | **M-SAN**** | **0.778 ± 0.021** | **0.589 ± 0.025** | | | M-SAN | 0.614 ± 0.010 | 0.437 ± 0.013 |
| | | M-HGAT | 0.721 ± 0.010 | 0.494 ± 0.016 | | | M-HGAT | 0.588 ± 0.011 | 0.437 ± 0.008 |
| | Biological Int. | B-GNN | 0.646 ± 0.018 | 0.415 ± 0.018 | | Biological Int. | B-GNN | 0.559 ± 0.017 | 0.371 ± 0.037 |
| | | M-GNN* | 0.710 ± 0.011 | 0.557 ± 0.008 | | | M-GNN | 0.587 ± 0.015 | 0.404 ± 0.028 |
| | | M-SAN* | 0.742 ± 0.007 | 0.582 ± 0.016 | | | M-SAN | 0.592 ± 0.009 | 0.433 ± 0.013 |
| | | M-HGAT | 0.689 ± 0.028 | 0.463 ± 0.027 | | | M-HGAT | 0.586 ± 0.017 | 0.441 ± 0.014 |
| **Colon Cancer** | | MLP | 0.489 ± 0.072 | 0.146 ± 0.032 | **Ovarian Cancer** | | MLP | 0.509 ± 0.037 | 0.481 ± 0.044 |
| | | GGNN | 0.638 | - | | | - | - | - |
| | Patient Sim. | B-GNN | 0.652 ± 0.022 | 0.303 ± 0.031 | | Patient Sim. | B-GNN | 0.592 ± 0.011 | 0.510 ± 0.017 |
| | | M-GNN | 0.611 ± 0.015 | 0.268 ± 0.020 | | | M-GNN | 0.624 ± 0.013 | 0.565 ± 0.016 |
| | | M-SAN | 0.657 ± 0.007 | 0.259 ± 0.007 | | | M-SAN | 0.660 ± 0.014 | 0.585 ± 0.018 |
| | | M-HGAT | 0.703 ± 0.009 | 0.306 ± 0.008 | | | M-HGAT | 0.631 ± 0.006 | 0.568 ± 0.022 |
| | Biological Int. | **B-GNN** | 0.711 ± 0.006 | **0.392 ± 0.025** | | Biological Int. | B-GNN | 0.608 ± 0.026 | 0.488 ± 0.092 |
| | | **M-GNN*** | **0.742 ± 0.019** | 0.310 ± 0.025 | | | **M-GNN*** | **0.672 ± 0.022** | 0.632 ± 0.024 |
| | | M-SAN | 0.664 ± 0.025 | 0.261 ± 0.019 | | | M-SAN | 0.569 ± 0.028 | 0.339 ± 0.101 |
| | | M-HGAT | 0.732 ± 0.012 | 0.302 ± 0.013 | | | **M-HGAT*** | 0.670 ± 0.005 | **0.649 ± 0.007** |

(Papillon et al., 2023; Feng et al., 2021). This supports the idea that architectures that reflect the underlying topology of the data lead to improved performance.

As we did not inject any biological knowledge into our models *a priori*, in this section, we evaluate whether our top performing models were capable of learning this for themselves. We identified the most important interactions in our TDL models by iteratively removing each of the higher order simplices/hyperedges and calculating the drop in ROC-AUC score that this produced. For the most important simplex/hyperedge, if the nodes were from the mRNA expression or DNA methylation networks, they were mapped to the genes that they represent, and if the nodes were from the mi-

Table 3: Averaged performance of models across all cancer types.

| | Model | Average ROC-AUC | Average F1 |
|---|---|---|---|
| | GGNN | 0.631 | - |
| Pat. Sim. | B-GNN | 0.626 | 0.414 |
| | M-GNN | 0.641 | 0.460 |
| | **M-SAN** | **0.665** | **0.482** |
| | M-HGAT | 0.646 | 0.454 |
| Bio. Int. | B-GNN | 0.615 | 0.421 |
| | M-GNN | 0.648 | 0.456 |
| | M-SAN | 0.618 | 0.387 |
| | **M-HGAT** | **0.672** | **0.497** |

croRNA expression network, they were first mapped to gene targets using the bioinformatics tool MirTarget[2] where the target score for the genes was 95 or higher. The ToppGene suite (Chen et al., 2009) was then used to analyze the functional enrichment of these genes. Table 4 shows the top 5 gene ontology (GO) biological process terms that are significantly enriched for the top gene list from each of the cancer types, together with the Bonferroni-adjusted p-values.

We see that the genes identified from the breast cancer and colon cancer models are associated with processes involved in tumor migration and invasion – key steps in the metastatic cascade (Friedl & Wolf, 2003). Interestingly, the breast cancer model also highlights bone development, as metastasis of breast cancer most commonly occurs to the skeleton (Akhtari et al., 2008). Neuron generation is additionally identified by the colon model – nerves can enhance cancer progression, and cancer cells can stimulate nerve growth, and this may play a particularly important role in colon cancer as the intestines contain a nervous system of their own (Schonkeren et al., 2021). The development of new blood vessels is also an indication of tumor aggressiveness (Nishida et al., 2006). Our model for kidney cancer (which is specifically renal clear cell carcinoma) identifies processes important for tube development, which is relevant as this type of cancer begins in the tubules of the kidneys. Additionally, the extent of cell differentiation reflects cancer aggression (Jögi et al., 2012). The genes identified from the melanoma, lung cancer and ovarian cancer models reflect that disregulation of transcription and translation are involved in cancer proliferation (Song et al., 2021; Zatzman et al., 2022). Additionally, ribonucleoproteins and mitochondrial tRNAs are specifically linked to the development and progression of lung cancer (Lu et al., 2022; Bian et al., 2021).

Table 11 in the Appendix displays the results for negative controls – for each cancer type, we randomly selected nodes from the same model and same omics type to make up a similar number of gene features as in Table 4. We observe that for breast cancer and lung cancer, there were no significantly enhanced (Bonferroni-adjusted p-value $< 0.05$) GO biological processes in these gene lists, and for ovarian cancer, there was only one significantly enhanced GO biological process. Other than general translation and transcription processes, there are few processes that are tied to cancer survival and progression, and in general, the p-values and overlap between the GO biological processes and the input gene list are lower, compared to Table 4 using the top simplex/hyperedge from each model.

## 5  CONCLUSIONS

In this paper, we demonstrate the utility of the Mapper algorithm and TDL models toward improving cancer survival predictions using multi-omics data. Multi-omics data offer a rich representation of the biological processes occurring within individual tumor samples, however, the vast number of protein-coding genes and possible transcriptional and post-transcriptional regulation processes involving these genes, makes finding the relevant biological interactions within the data a challenging task. In this work, we built upon the success of GNN-based models utilizing patient similarity and biological interaction networks, using topological tools to refine and extend the representations of these networks. Node pooling using the Mapper algorithm helped to sparsify and reduce the networks, leading to improved performance and faster training particularly for the large biologi-

---

[2]https://mirdb.org/

Table 4: The top 5 GO biological processes significantly enriched for the genes from the most important simplex/hyperedge from the top-performing model for each cancer type. The Bonferroni-adjusted p-value, the total number of genes in the GO annotation, the total number of genes identified by the model, and the number of genes in both are shown.

| Cancer | GO: Biological Process | | Number of genes | | | |
|---|---|---|---|---|---|---|
| | ID | Name | Adj. p-value | Tot. GO BP | Tot. model | Both |
| Breast | GO:0030334 | regulation of cell migration | $2.14 \times 10^{-5}$ | 1211 | 121 | 26 |
| (mRNA) | GO:2000145 | regulation of cell motility | $6.65 \times 10^{-5}$ | 1280 | 121 | 26 |
| | GO:0040012 | regulation of locomotion | $1.38 \times 10^{-4}$ | 1327 | 121 | 26 |
| | GO:0060348 | bone development | $1.67 \times 10^{-4}$ | 264 | 121 | 12 |
| | GO:0040011 | locomotion | $5.99 \times 10^{-4}$ | 1529 | 121 | 27 |
| Colon | GO:0007155 | cell adhesion | $3.11 \times 10^{-43}$ | 1675 | 1884 | 336 |
| (mRNA) | GO:0007267 | cell-cell signaling | $1.02 \times 10^{-39}$ | 1584 | 1884 | 316 |
| | GO:0048699 | generation of neurons | $3.11 \times 10^{-37}$ | 1858 | 1884 | 346 |
| | GO:0072359 | circulatory system development | $7.30 \times 10^{-35}$ | 1442 | 1884 | 286 |
| | GO:0016477 | cell migration | $3.64 \times 10^{-34}$ | 1847 | 1884 | 337 |
| Kidney | GO:0035295 | tube development | $1.58 \times 10^{-5}$ | 1402 | 172 | 35 |
| (miRNA) | GO:0035239 | tube morphogenesis | $5.69 \times 10^{-5}$ | 1125 | 172 | 30 |
| | GO:0045595 | regulation of cell differentiation | $1.15 \times 10^{-4}$ | 1974 | 172 | 41 |
| | GO:0045597 | + regulation of cell differentiation | $2.89 \times 10^{-4}$ | 1141 | 172 | 29 |
| | GO:0048646 | anatomical structure formation involved in morphogenesis | $7.28 \times 10^{-4}$ | 1483 | 172 | 33 |
| Melanoma | GO:0045893 | + regulation of DNA-templated transcription | $2.41 \times 10^{-8}$ | 1824 | 346 | 72 |
| (miRNA) | GO:1902680 | + regulation of RNA biosynthetic process | $2.61 \times 10^{-8}$ | 1827 | 346 | 72 |
| | GO:0045944 | + regulation of transcription by RNA polymerase II | $4.24 \times 10^{-6}$ | 1390 | 346 | 56 |
| | GO:0045892 | - regulation of DNA-templated transcription | $1.47 \times 10^{-5}$ | 1399 | 346 | 55 |
| | GO:1902679 | - regulation of RNA biosynthetic process | $2.09 \times 10^{-5}$ | 1413 | 346 | 55 |
| Lung | GO:0002181 | cytoplasmic translation | $4.95 \times 10^{-44}$ | 172 | 1471 | 82 |
| (mRNA) | GO:0006412 | translation | $2.25 \times 10^{-42}$ | 824 | 1471 | 184 |
| | GO:0022613 | ribonucleoprotein complex biogenesis | $5.58 \times 10^{-32}$ | 515 | 1471 | 126 |
| | GO:0032543 | mitochondrial translation | $9.36 \times 10^{-30}$ | 133 | 1471 | 60 |
| | GO:0140053 | mitochondrial gene expression | $6.00 \times 10^{-29}$ | 176 | 1471 | 68 |
| Ovarian | GO:0045892 | - regulation of DNA-templated transcription | $1.22 \times 10^{-2}$ | 1399 | 188 | 31 |
| (miRNA) | GO:1902679 | - regulation of RNA biosynthetic process | $1.49 \times 10^{-2}$ | 1413 | 188 | 31 |
| | GO:0045893 | + regulation of DNA-templated transcription | $2.49 \times 10^{-2}$ | 1824 | 188 | 36 |
| | GO:1902680 | + regulation of RNA biosynthetic process | $2.59 \times 10^{-2}$ | 1827 | 188 | 36 |
| | GO:0051253 | - regulation of RNA metabolic process | $2.78 \times 10^{-2}$ | 1531 | 188 | 32 |

cal interaction networks. TDL models, including a new SAN architecture that performs message passing between cells of adjacent ranks, enabled the modeling of higher order interactions between nodes, which better represents real-life networks where pairwise interactions may not capture the full picture. We observed a particular improvement in performance in the HyperGAT model applied to the biological interaction network, which is likely due to its accurate representation of the underlying data, as biological interactions lend themselves naturally to a hypergraph representation.

Unlike previous approaches which relied on prior knowledge of biological interactions, our models are not limited by the current state of biological understanding, and have the potential to learn tissue and cancer specific interactions. Despite the lack of biological priors, our models were able to learn key interactions related to cancer progression and metastasis, as well as interactions linked specifically to the development or progression of each cancer type. Our results show that careful representation of the network extracted from complex and high-dimensional multi-omics data allowed our models to outperform models utilizing prior biological knowledge, highlighting the potential of topological methods for improving representations of real-life networks for increased performance.

## REPRODUCIBILITY STATEMENT

To ensure reproducible results, we have made our project code available at `https://anonymous.4open.science/r/topo-omics-061A/`, and included written descriptions within the Methods and Appendix sections. The raw data used in our project is publicly available at `http://acgt.cs.tau.ac.il/multi_omic_benchmark/download.html`, and our project code includes our full data processing pipeline, with the main details described in the Methods section. Our implementation of the Mapper algorithm is described in the Methods section and Appendix A.1, and our model architectures are described in the Methods section, and Appendices A.2 and A.3. The code for our models is extensively commented, and contains references to papers for each of the modules. Full training and hyperparameter details are provided in Appendix A.5, together with the hardware used. Software packages and versions are described in the requirements.txt file included in our code repository.

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

# A APPENDIX

## A.1 MAPPER NETWORK CONSTRUCTION DETAILS

The Mapper algorithm is an unsupervised algorithm commonly used to visualize high-dimensional data by representing it as a graph in a low-dimensional space through four steps: applying a filter function, constructing a covering, constructing the inverse covering, and clustering of the preimage Lee & Jung (2023); Hajij et al. (2022); Singh et al. (2007). The data are first represented as a point cloud, and the filter function is a continuous map that transforms the data into a lower-dimensional space, where a covering is constructed for the data such that the covers overlap. Using the inverse of the filter function, the covering is mapped back into the high-dimensional space, and the data points within each inverse cover is clustered, forming the nodes of the Mapper graph.

We used PCA with 3 principle components as the filter function, and DBSCAN for the clustering algorithm. We used a cubical cover with a number of intervals between 5 and 20 (which is tuned as a hyperparameter) and overlap fraction of 0.1 for the biological interaction network. For the patient similarity network, we used 10 intervals for mRNA expression and DNA methylation data, and 20 intervals for microRNA expression data for reduced compaction of the graph, and $10^{-4}$ for the overlap fraction to reduce the probability of one patient being grouped into more than one node.

Once the node pooling has been obtained, the features for each Mapper node are calculated as the average of the features of the nodes combined to form the Mapper node. The distance correlation (which also captures non-linear relationships (Székely et al., 2007)) between the Mapper nodes is used to construct the graphs, simplicial complexes and hypergraphs. A threshold for the distance correlation is calculated from the average degree of each node, which is tuned as a hyperparameter, and nodes within this distance correlation of each other are connected via an edge. In simplicial complexes and hypergraphs, if three or four nodes are fully connected, they are formed into higher order simplexes or hyperedges.

## A.2 MAPPER SAN MODEL DETAILS

The message passing between SAN layers is described in the main text. There are two SAN layers in each omics-specific model, with slight differences depending on whether the model used a patient similarity network or biological interaction network. In the case of the patient similarity Mapper SAN, the two SAN layers transform the feature space into 400 and 200 dimensions respectively. A linear layer is first applied to transform the node features into the binary survival classes, and this output is then multiplied on the left by the deterministic mapping from Mapper nodes to patients, to obtain the patient-level label probabilities. In the case of the biological interaction Mapper SAN, the two SAN layers transform the feature space into 4 and 2 dimensions respectively. A global mean pooling layer aggregates the output node features, and a final linear layer transforms this output into the binary survival classes.

The outputs from the omics-specific models are combined into a View Correlation Discovery Network (introduced in Wang et al. (2019) and adapted for multi-omics in Wang et al. (2021)) which integrates the predictions, learns label correlations between the omics outputs, and produces the final label prediction.

## A.3 MAPPER HYPERGAT MODEL DETAILS

From the Mapper network, for each omics type for each patient, we have the feature matrix $X \in \mathbb{R}^{K \times 1}$ where $K$ is the number of Mapper-reduced features, and the incidence matrix $A \in \mathbb{R}^{K \times E}$ where $E$ is the number of hyperedges, with entries being 1 if a node is part of a hyperedge, and 0 otherwise.

As in Ding et al. (2020), we apply node-level attention followed by edge-level attention. The node-level attention coefficient, $\alpha_{ix}$ of node $v_x$ in the hyperedge $e_i$ is given by:

$$\alpha_{ix} = \frac{\exp(a_1^T u_x)}{\sum_{v_p \in e_i} \exp(a_1^T u_p)} \tag{4}$$
$$u = \text{LeakyRELU}(W^1 X)$$

where $W^1 \in \mathbb{R}^{K \times h}$ (where $h$ is the number of features in the next layer) is a weight matrix, and $a_1 \in \mathbb{R}^h$ are the attention weights. The calculated node-level attention values are then used to compute a representation for the hyperedges:

$$f_i = \sigma\Big( \sum_{v_x \in e_i} \alpha_{ix} W^1 X \Big) \tag{5}$$

where $\sigma$ indicates the softmax function. The edge-level attention coefficients are then calculated as:

$$\beta_{jy} = \frac{\exp(a_2^T w_i)}{\sum_{e_p} \exp(a_2^T w_p)} \tag{6}$$
$$w_i = \text{LeakyRELU}(W_i^2 f_i || W_y^1 v_y)$$

where $W^2 \in \mathbb{R}^{h \times h}$ and $a_2 \in \mathbb{R}^{2h}$ are additional weight matrix/vector. These are used to calculate the node features of the next layer as:

$$v_y' = \sigma\Big( \sum_{e_j} \beta_{jy} W_2 f_j \Big) \tag{7}$$

Similar to the Mapper SAN, we use two HyperGAT layers for each omics specific model, with the patient similarity HyperGAT transforming the feature space into 200 and 100 dimensions respectively, before applying a linear layer followed by the node mapping to obtain binary classification probabilities for each node. For the biological interaction HyperGAT, the two layers transform the feature space into 4 and 2 dimensions respectively, before applying a global mean pooling followed by a final linear layer. The outputs from the omics-specific models are passed into the downstream VCDN exactly as in the GNNs and Mapper SANs.

A.4 MEDIAN SURVIVAL TIMES BY CANCER TYPE

The median survival times for each cancer type used to dichotomize survival times are shown in Table 5. Median survival times are calculated using deceased patients from the entire dataset as data are dichotomized prior to the stratified train-test split.

Table 5: Median survival patients for each cancer type

| Cancer type | Median survival (days) |
| --- | --- |
| Breast | 1324 |
| Colon | 426 |
| Kidney | 806 |
| Melanoma | 1093 |
| Lung | 559 |
| Ovarian | 1078 |

A.5 HYPERPARAMETER TUNING AND TRAINING DETAILS

Following the method of Wang et al. (2021), we pretrain the omics specific models before combining them into the full model. We tune the hyperparameters using loss computed on the validation set

in three stages: first we find the optimal number of epochs, learning rate and weight decay for the pretraining stage, then we optimize the learning rate, weight decay, dropout and batch size (for biological interaction models) for the full model, and finally, as the network structure may be different for each cancer type, we optimize the average degree for each of the omics networks individually for each cancer type, and for biological interaction networks, we also optimize the extent of compaction of the networks through the hyperparameter specifying the number of intervals for the covering used by the Mapper algorithm. This step-wise process for hyperparameter tuning reduces the risk of overfitting by constraining the hyperparameters which are not specific to the omics network for each cancer to reduce the number of hyperparameters tuned individually for each cancer subtype. However, the network specific hyperparameters must be tuned separately for each cancer subtype, as the connectivity of each network is different, such that they have differing propensities for overfitting and different optimal degree distributions.

Hyperparameter tuning is carried out using a grid search using the Weights & Biases[3] platform, and hyperparameter importance and correlation with loss evaluated on the test set are shown in Figures 4-9 for each of the models we introduced. For models where #intervals or max rank are hyperparameters but not shown, this is due to a separate sweep being set up for each value of these hyperparameters in order to limit the compute time for each sweep. The figures show the hyperparameter importance corresponding to the #intervals and max rank of the top model. It can be seen that runtime is frequently one of the top parameters in terms of importance, suggesting that the hyperparameter landscape is fairly flat for at least some of the omics networks in all the cancers.

Additionally, while we see some signs of overfitting for some hyperparameter combinations (runtime sometimes positively correlated with test loss), Figure 10 shows the validation loss during training for the top models for each cancer type, demonstrating that this does not occur for the optimal choice of hyperparameters.

Tables 6 and 8 show the pretraining and training hyperparameters for all the models in the paper. We note that the values for the average degree of the mRNA, DNA methylation and miRNA networks were chosen for a median degree of approximately 5-10 across all nodes, and due to the different levels of compaction in the biological interaction networks (governed by the #intervals parameter), the average degree values can appear very different even if the resulting degree distributions are similar.

Table 6: Hyperparameters for pretraining and full model

| Model | | Pre-train hyperparameters | | | Full model hyperparameters | | | | |
|---|---|---|---|---|---|---|---|---|---|
| | | Learning rate | Weight decay | Epochs | Learning rate | Weight decay | Dropout | Epochs | Batch |
| Patient sim. | B-GNN | $1 \times 10^{-3}$ | 0 | 500 | $5 \times 10^{-4}$ | 0 | 0.5 | 3500 | - |
| | M-GNN | $1 \times 10^{-3}$ | 0 | 1000 | $5 \times 10^{-4}$ | 0 | 0.5 | 7000 | - |
| | M-SAN | $1 \times 10^{-3}$ | $1 \times 10^{-3}$ | 500 | $5 \times 10^{-4}$ | $1 \times 10^{-3}$ | 0.5 | 7000 | - |
| | M-HGAT | $1 \times 10^{-3}$ | $1 \times 10^{-3}$ | 200 | $1 \times 10^{-4}$ | 0 | 0.6 | 7000 | - |
| Biological int. | B-GNN | $1 \times 10^{-3}$ | 0 | 500 | $1 \times 10^{-3}$ | 0 | 0.5 | 3500 | 32 |
| | M-GNN | $1 \times 10^{-3}$ | 0 | 500 | $1 \times 10^{-3}$ | 0 | 0.4 | 7000 | 32 |
| | M-SAN | $1 \times 10^{-3}$ | $1 \times 10^{-3}$ | 100 | $1 \times 10^{-3}$ | 0 | 0.5 | 7000 | 60 |
| | M-HGAT | $5 \times 10^{-4}$ | $1 \times 10^{-3}$ | 100 | $1 \times 10^{-4}$ | $1 \times 10^{-3}$ | 0.6 | 7000 | 32 |

The patient similarity benchmark GNN took on average 2 min to train one model on a 14 core MacBook Pro M4 with 24GB memory. The remaining models were trained on an Nvidia A100 GPU with 80GB memory on a Linux high performance computing cluster, and Table 7 shows the average time required for training, and the number of parameters for each model.

---

[3] https://wandb.ai/site/

Table 7: Time to train models and model size comparisons. For the patient similarity benchmark GNN, an asterisk indicates that the model was trained locally on a MacBook Pro, whereas other models were trained on an Nvidia GPU (see text).

| Model | | Average time to train model | Number of parameters |
|---|---|---|---|
| Patient sim. | B-GNN | 2 min* | 21 M |
| | M-GNN | 4 min | 21 M |
| | M-SAN | 10 min | 102 M |
| | M-HGAT | 7 min | 5 M |
| Biological int. | B-GNN | 45 min | 414 |
| | M-GNN | 12 – 25 min | 414 |
| | M-SAN | 1.5 – 3.5 hours | 588 |
| | M-HGAT | 0.5 – 2.5 hours | 276 |

## A.6 GNNEXPLAINER ANALYSIS OF PATIENT SIMILARITY MODELS

Table 8: Cancer type specific hyperparameters

| Cancer | Hyperparameters | Patient similarity | | | | Biological interaction | | | |
|---|---|---|---|---|---|---|---|---|---|
| | | B-GNN | M-GNN | M-SAN | M-HGAT | B-GNN | M-GNN | M-SAN | M-HGAT |
| Breast | mRNA | 0.4 | 2.0 | 1.6 | 1.2 | 0.1 | 0.6 | 6.0 | 0.8 |
| | DNA Meth | 1.4 | 0.6 | 0.6 | 0.6 | 2.0 | 1.0 | 10.0 | 0.8 |
| | miRNA | 1.6 | 2.8 | 2.8 | 2.2 | 2.0 | 2.4 | 12.0 | 2.6 |
| | #intervals | - | - | - | - | - | 15 | 5 | 15 |
| | max rank | - | - | 2 | - | - | - | 2 | - |
| Colon | mRNA | 0.4 | 1.2 | 2.8 | 2.8 | 2.0 | 6.0 | 4.0 | 10.0 |
| | DNA Meth | 1.0 | 1.4 | 0.8 | 1.4 | 3.0 | 4.0 | 12.0 | 10.0 |
| | miRNA | 2.0 | 2.0 | 2.4 | 2.6 | 3.0 | 12.0 | 6.0 | 6.0 |
| | #intervals | - | - | - | - | - | 5 | 5 | 5 |
| | max rank | - | - | 2 | - | - | - | 2 | - |
| Kidney | mRNA | 0.4 | 2.0 | 2.4 | 1.2 | 1.0 | 8.0 | 8.0 | 0.8 |
| | DNA Meth | 1.0 | 1.0 | 1.0 | 1.0 | 2.0 | 0.6 | 10.0 | 0.6 |
| | miRNA | 2.0 | 2.8 | 2.6 | 2.2 | 4.0 | 1.6 | 10.0 | 2.8 |
| | #intervals | - | - | - | - | - | 20 | 5 | 15 |
| | max rank | - | - | 3 | - | - | - | 2 | - |
| Melanoma | mRNA | 2.0 | 2.0 | 2.0 | 1.2 | 1.0 | 8.0 | 3.0 | 2.0 |
| | DNA Meth | 1.0 | 1.0 | 0.6 | 0.8 | 3.0 | 8.0 | 1.0 | 4.0 |
| | miRNA | 2.4 | 2.4 | 2.2 | 2.8 | 4.0 | 10.0 | 2.0 | 3.0 |
| | #intervals | - | - | - | - | - | 5 | 10 | 10 |
| | max rank | - | - | 3 | - | - | - | 2 | - |
| Lung | mRNA | 2.0 | 1.2 | 2.4 | 2.4 | 0.2 | 1.0 | 4.0 | 4.0 |
| | DNA Meth | 1.4 | 1.2 | 1.4 | 1.4 | 2.0 | 1.2 | 4.0 | 8.0 |
| | miRNA | 1.6 | 2.0 | 2.8 | 2.2 | 1.0 | 2.4 | 10.0 | 4.0 |
| | #intervals | - | - | - | - | - | 15 | 5 | 5 |
| | max rank | - | - | 3 | - | - | - | 2 | - |
| Ovarian | mRNA | 1.2 | 1.6 | 2.0 | 1.2 | 1.0 | 0.2 | 10.0 | 1.0 |
| | DNA Meth | 0.6 | 1.2 | 1.4 | 1.4 | 1.0 | 0.2 | 6.0 | 1.4 |
| | miRNA | 2.4 | 2.4 | 2.2 | 2.4 | 2.0 | 1.4 | 6.0 | 2.4 |
| | #intervals | - | - | - | - | - | 20 | 5 | 15 |
| | max rank | - | - | 2 | - | - | - | 2 | - |

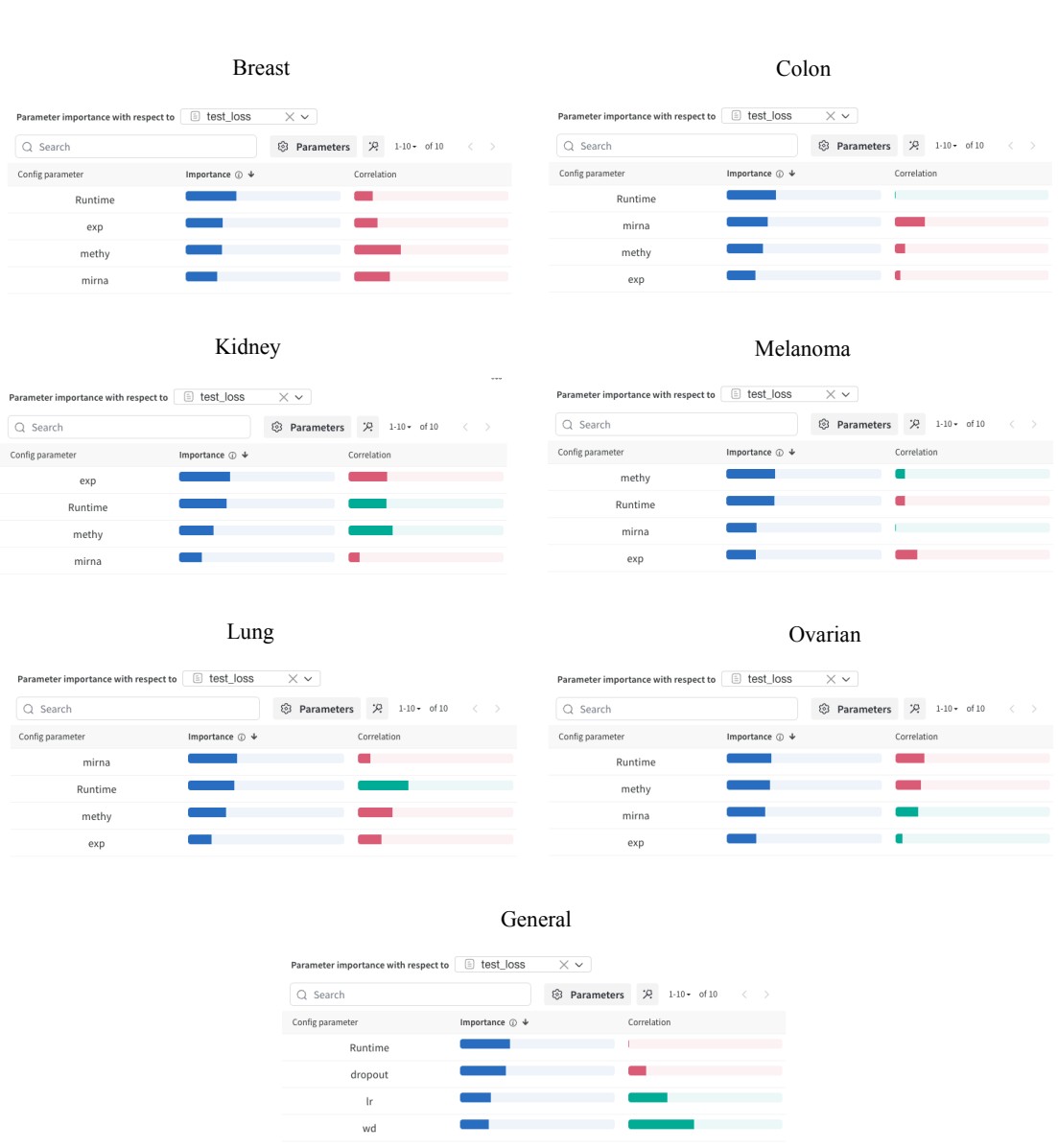

Figure 4: Hyperparameter sensitivity and correlations for patient similarity M-GNN models.

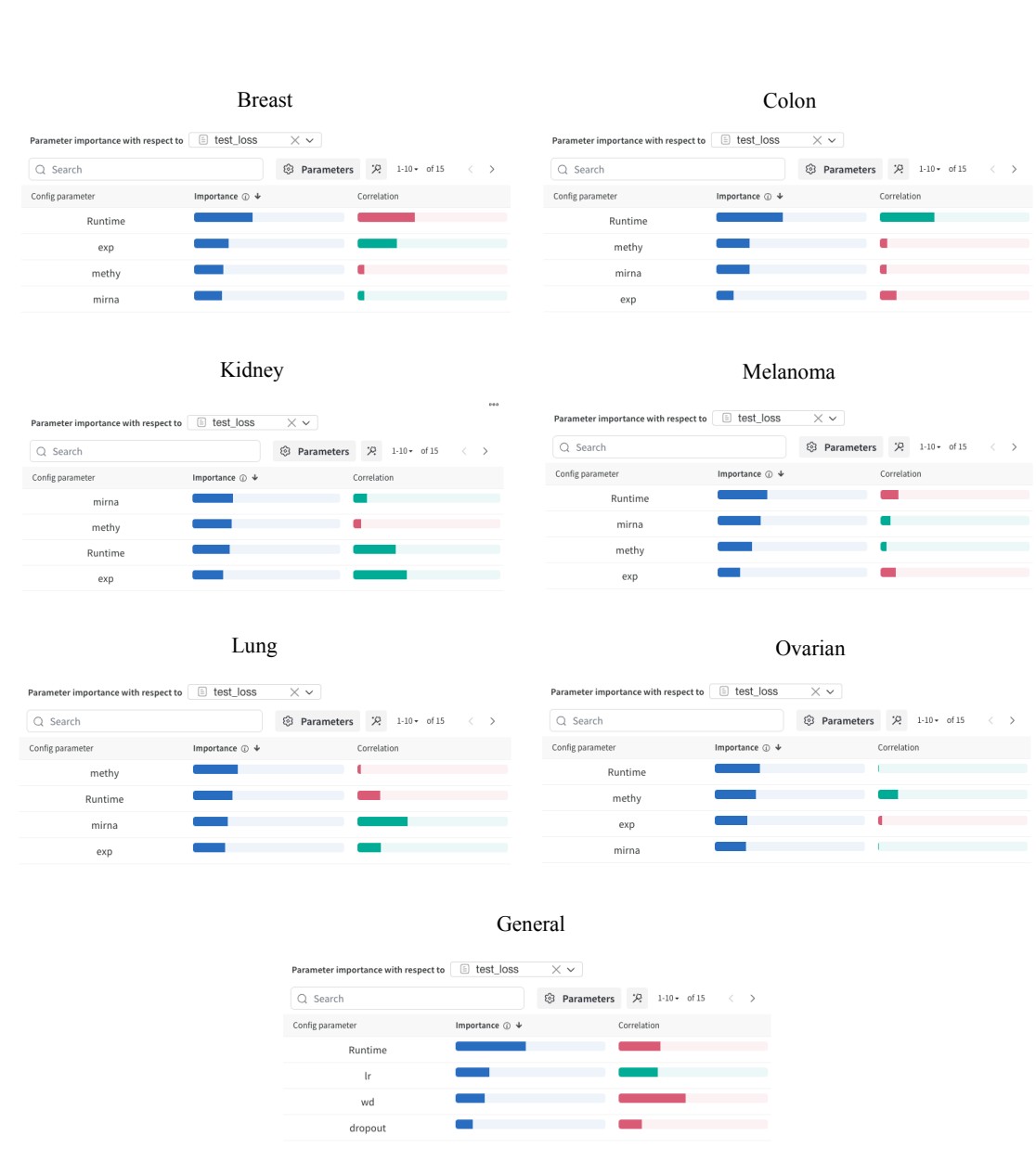

Figure 5: Hyperparameter sensitivity and correlations for biological interaction M-GNN models.

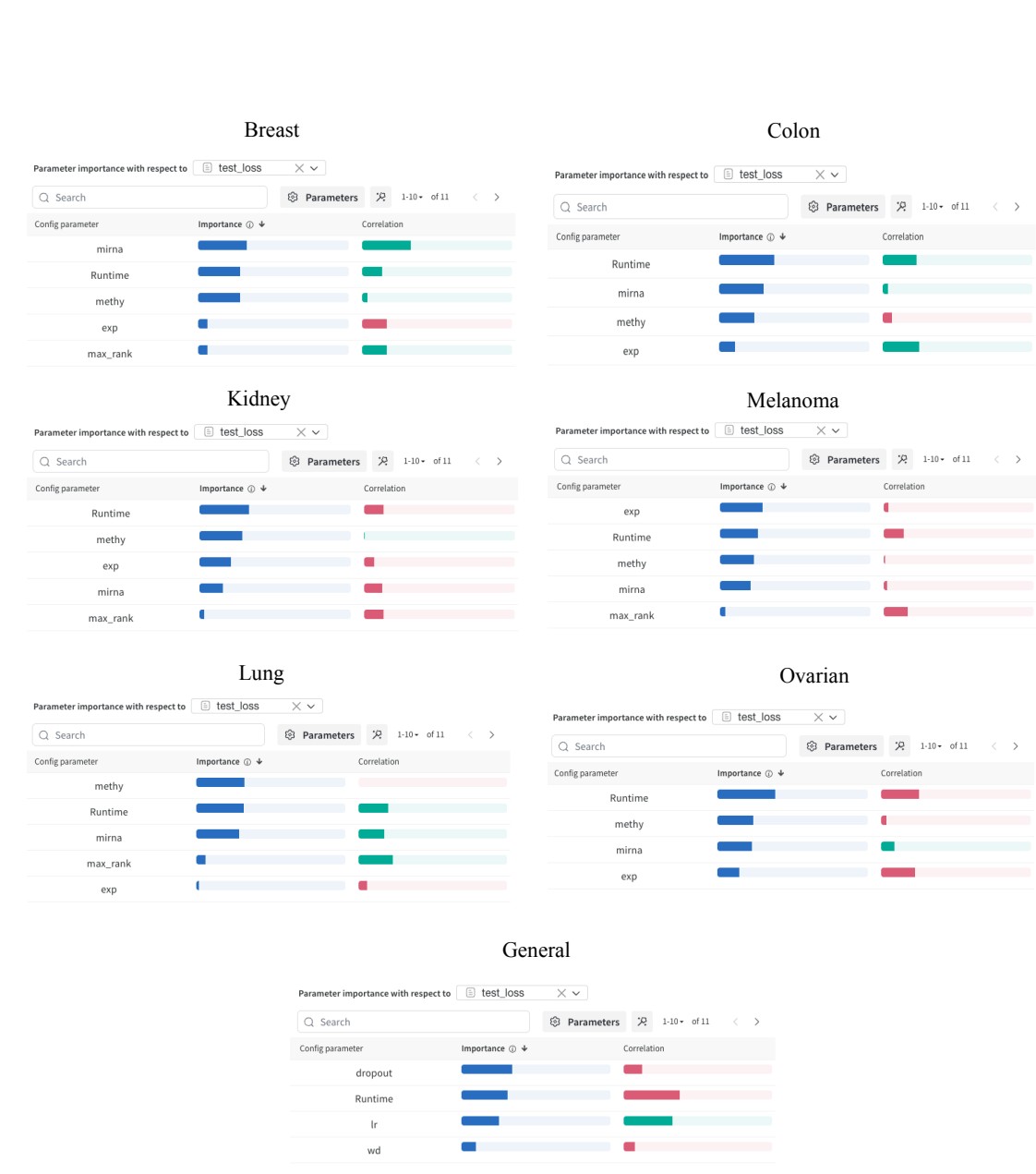

Figure 6: Hyperparameter sensitivity and correlations for patient similarity M-SAN models.

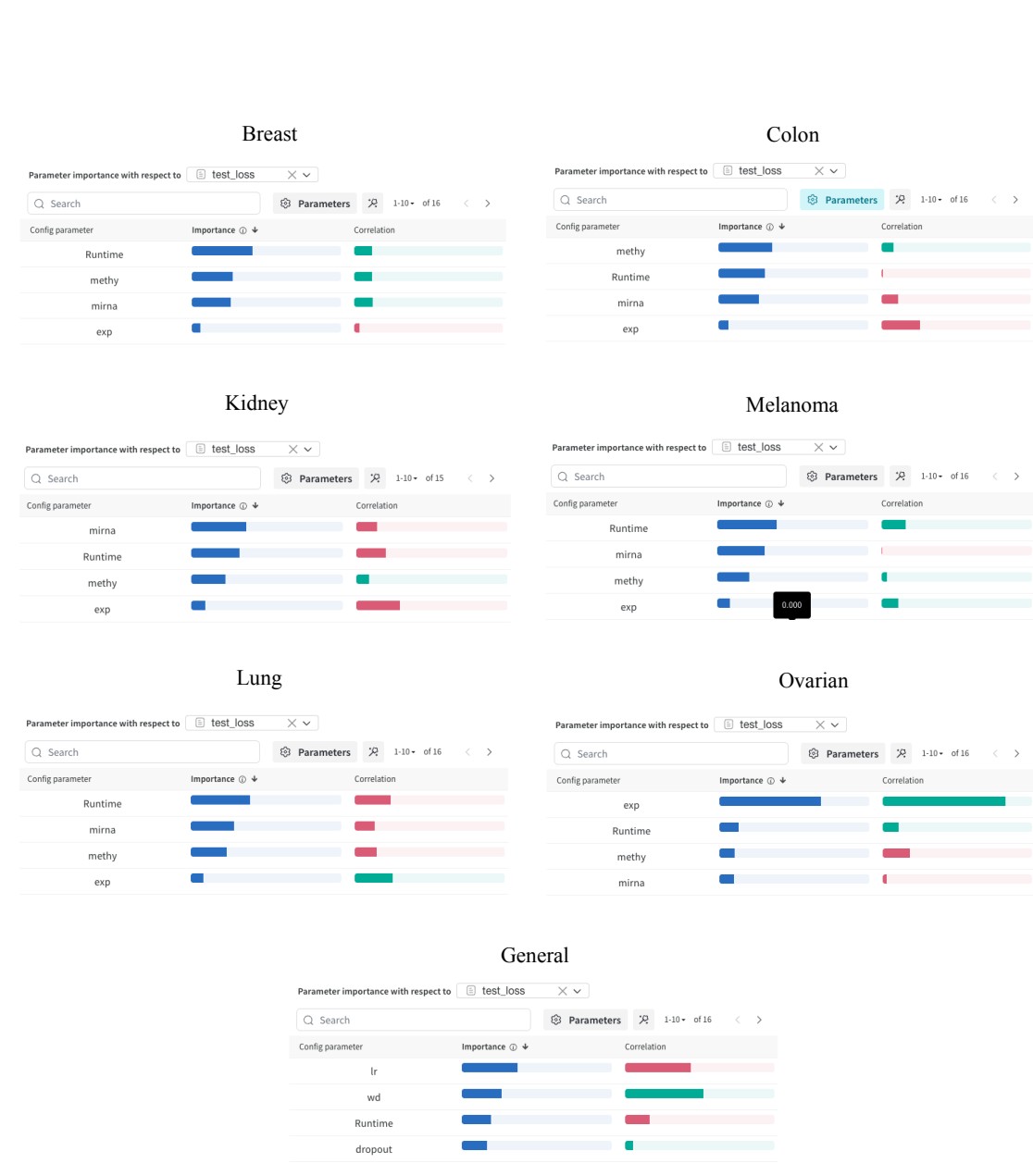

Figure 7: Hyperparameter sensitivity and correlations for biological interaction M-SAN models.

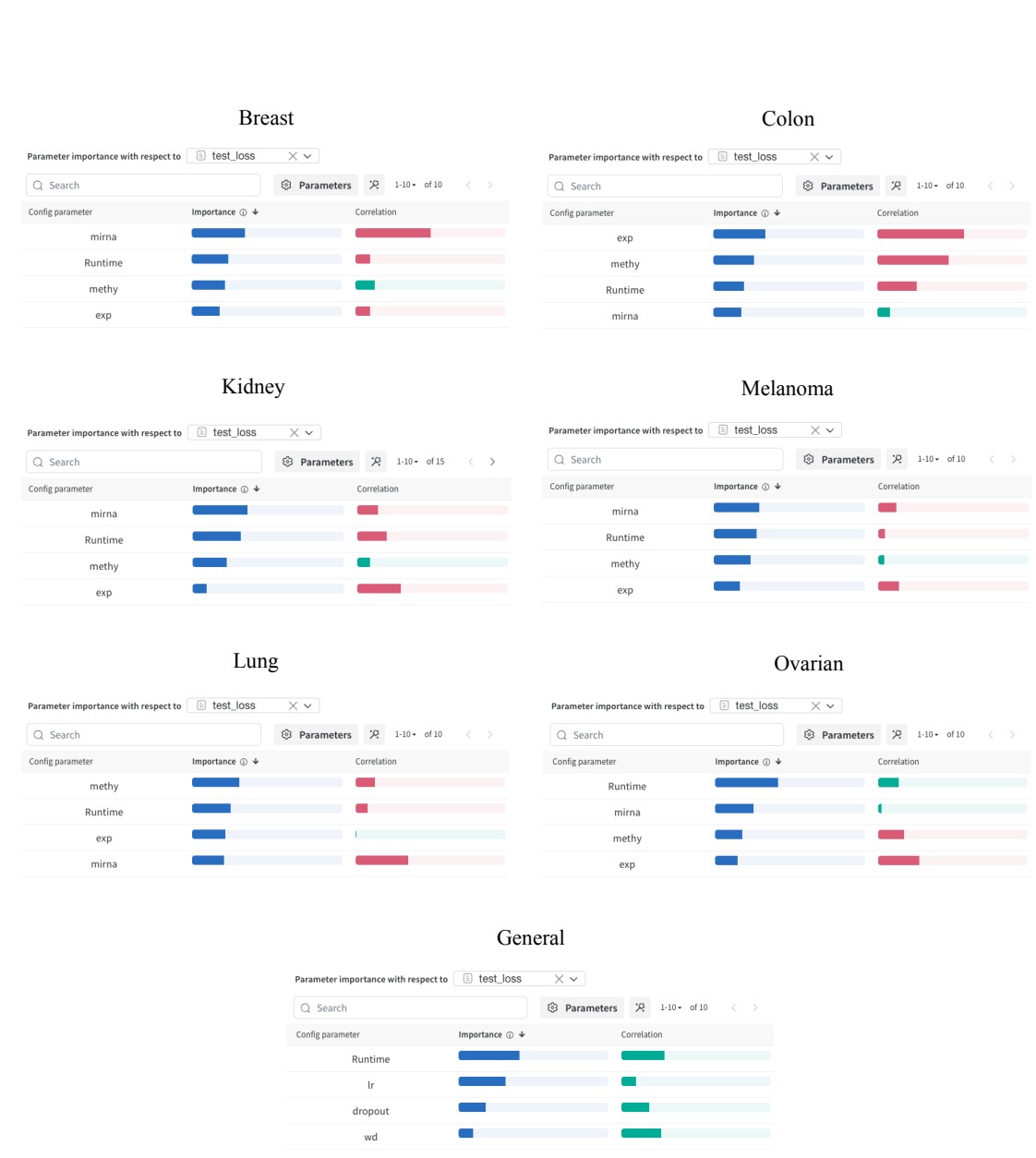

Figure 8: Hyperparameter sensitivity and correlations for patient similarity M-HGAT models.

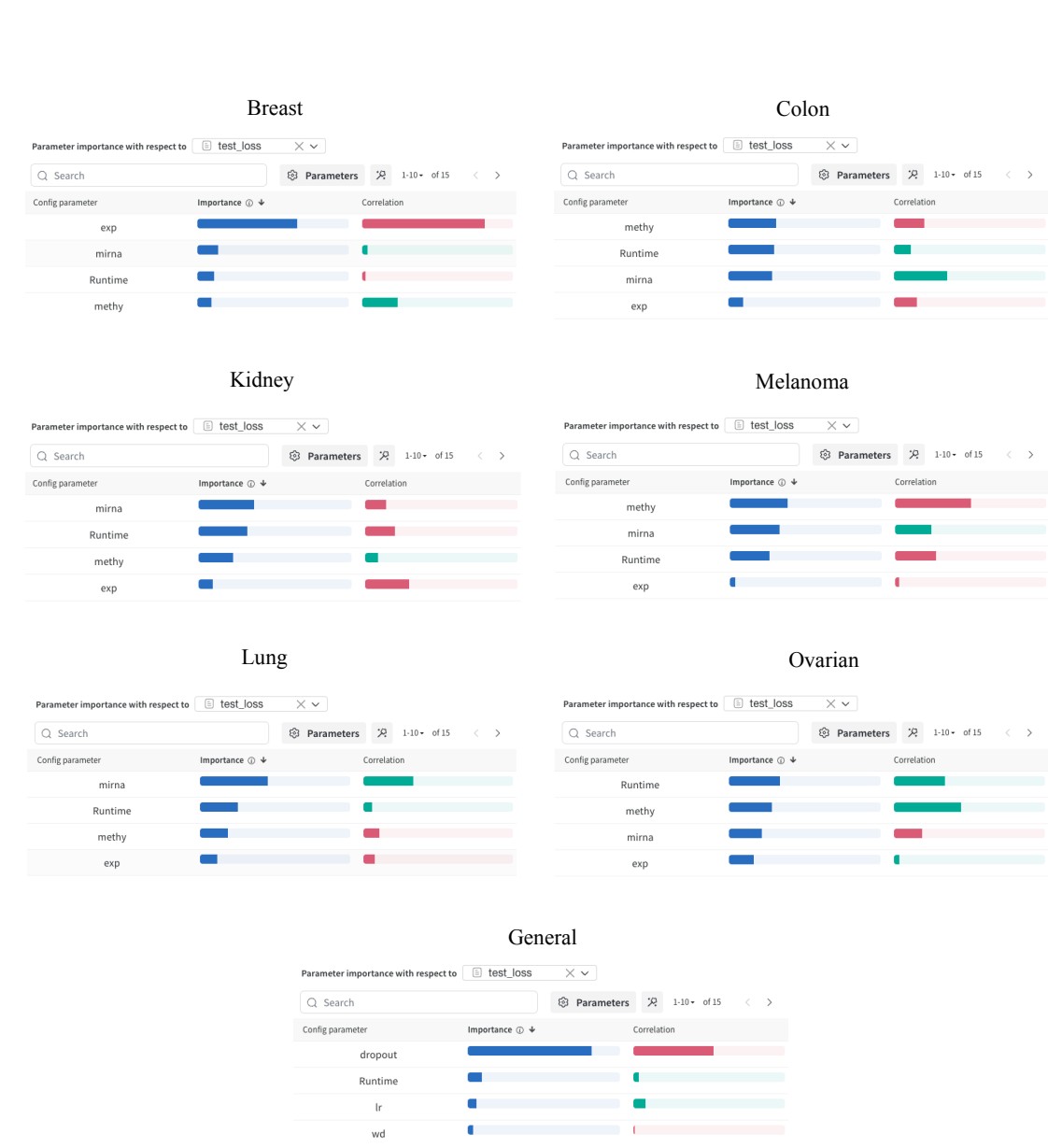

Figure 9: Hyperparameter sensitivity and correlations for biological interaction M-GAT models.

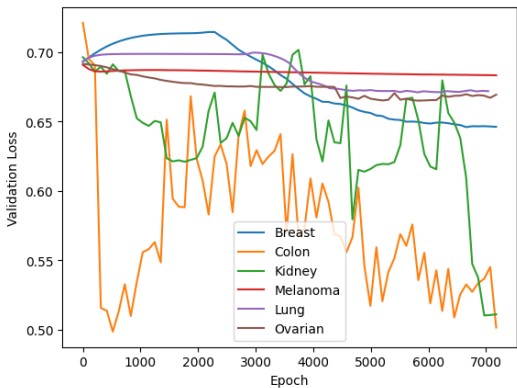

Figure 10: Validation loss curves for top-performing models for all cancer types.

Table 9: Fraction of patients in each omics graph that are important for correct label prediction, and grouped together in the same Mapper node. The mean, minimum, median, and maximum are computed as different methods for aggregating across the omics types, and the average of each aggregation method across the Mapper and Benchmark groups are compared at the bottom of the table.

| Group | Cancer | mRNA | DNA Meth | miRNA | Mean | Min | Median | Max |
|---|---|---|---|---|---|---|---|---|
| Mapper | Kidney | 0.174 | 0.114 | 0.141 | 0.143 | 0.114 | 0.141 | 0.174 |
| | Lung | 0.071 | 0.071 | 0.037 | 0.060 | 0.037 | 0.071 | 0.071 |
| | Ovarian | 0.066 | 0.056 | 0.042 | 0.055 | 0.042 | 0.056 | 0.066 |
| Benchmark | Breast | 0.065 | 0.101 | 0.040 | 0.069 | 0.040 | 0.065 | 0.101 |
| | Colon | 0.082 | 0.041 | 0.035 | 0.053 | 0.035 | 0.041 | 0.082 |
| | Melanoma | 0.102 | 0.036 | 0.099 | 0.079 | 0.036 | 0.099 | 0.102 |
| | **Average over Mapper group** | | | | **0.086** | **0.065** | **0.089** | **0.104** |
| | **Average over Benchmark group** | | | | **0.067** | **0.037** | **0.068** | **0.095** |

Table 10: Fraction of higher order simplices/hyperedges that contain nodes important for correct label prediction. The mean, minimum, median, and maximum are computed as different methods for aggregating across the omics types, and the average of each aggregation method across the TDL and GNN groups are compared at the bottom of the table.

| Group | Cancer | mRNA | DNA Meth | miRNA | Mean | Min | Median | Max |
|---|---|---|---|---|---|---|---|---|
| TDL | Breast | 0.525 | 0.633 | 0.696 | 0.618 | 0.525 | 0.633 | 0.696 |
| | Colon | 0.276 | 0.863 | 0.988 | 0.709 | 0.276 | 0.863 | 0.988 |
| | Ovarian | 0.364 | 0.842 | 0.684 | 0.630 | 0.364 | 0.684 | 0.842 |
| GNN | Kidney | 0.571 | 0.243 | 0.914 | 0.576 | 0.243 | 0.571 | 0.914 |
| | Melanoma | 0.265 | 0.547 | 0.711 | 0.508 | 0.265 | 0.547 | 0.711 |
| | Lung | 0.102 | 0.549 | 0.583 | 0.411 | 0.102 | 0.549 | 0.583 |
| | **Average over TDL group** | | | | **0.652** | **0.388** | **0.727** | **0.842** |
| | **Average over GNN group** | | | | **0.498** | **0.203** | **0.556** | **0.736** |

Table 11: The top 5 GO biological processes significantly enriched for the genes from a randomly-selected set of features from the same model and omics type as in Table 4. The Bonferroni-adjusted p-value, the total number of genes in the GO annotation, the total number of genes identified by the model, and the number of genes in both are shown.

| Cancer | GO: Biological Process | | Number of genes | | | |
|--------|----|-----|------|------|------|------|
| | ID | Name | Adj. p-value | Tot. GO BP | Tot. model | Both |
| Breast | - | - | - | - | 111 | |
| (mRNA) | - | - | - | - | 111 | - |
| | - | - | - | - | 111 | - |
| | - | - | - | - | 111 | - |
| | - | - | - | - | 111 | - |
| Colon | GO:0048285 | organelle fission | $3.60 \times 10^{-6}$ | 579 | 1038 | 66 |
| (mRNA) | GO:0032543 | mitochondrial translation | $4.03 \times 10^{-6}$ | 134 | 1038 | 27 |
| | GO:0000280 | nuclear division | $1.22 \times 10^{-5}$ | 518 | 1038 | 60 |
| | GO:0006520 | amino acid metabolic process | $4.83 \times 10^{-5}$ | 324 | 1038 | 43 |
| | GO:0043436 | oxoacid metabolic process | $9.57 \times 10^{-5}$ | 1056 | 1038 | 96 |
| Kidney | GO:0045893 | + regulation of DNA-templated transcription | $1.69 \times 10^{-3}$ | 1826 | 134 | 31 |
| (miRNA) | GO:1902680 | + regulation of RNA biosynthetic process | $1.73 \times 10^{-3}$ | 1828 | 134 | 31 |
| | GO:0045663 | + regulation of myoblast differentiation | $4.17 \times 10^{-3}$ | 54 | 134 | 6 |
| | GO:0045661 | regulation of myoblast differentiation | $7.18 \times 10^{-3}$ | 92 | 134 | 7 |
| | GO:0045944 | + regulation of transcription by RNA poly. II | $8.66 \times 10^{-3}$ | 1382 | 134 | 25 |
| Melanoma | GO:0045893 | + regulation of DNA-templated transcription | $3.94 \times 10^{-5}$ | 1826 | 482 | 82 |
| (miRNA) | GO:1902680 | + regulation of RNA biosynthetic process | $4.14 \times 10^{-5}$ | 1828 | 482 | 82 |
| | GO:0031175 | + neuron projection development | $2.46 \times 10^{-3}$ | 1304 | 482 | 60 |
| | GO:0045944 | + regulation of transcription by RNA poly. II | $7.79 \times 10^{-3}$ | 1382 | 482 | 61 |
| | GO:0048699 | generation of neurons | $1.30 \times 10^{-2}$ | 1892 | 482 | 76 |
| Lung | - | - | - | - | 1390 | - |
| (mRNA) | - | - | - | - | 1390 | - |
| | - | - | - | - | 1390 | - |
| | - | - | - | - | 1390 | - |
| | - | - | - | - | 1390 | - |
| Ovarian | GO:0035239 | tube morphogenesis | $3.65 \times 10^{-2}$ | 1147 | 162 | 24 |
| (miRNA) | - | - | - | - | 162 | - |
| | - | - | - | - | 162 | - |
| | - | - | - | - | 162 | - |
| | - | - | - | - | 162 | - |

