# OpenReview forum: "Topology over biology: network representation improves multi-omics models without need for prior knowledge"
_ICLR.cc/2026/Conference — Submitted to ICLR 2026_

### Official Review · Reviewer_8VSi · 2025-10-23

**Soundness:** 2
**Presentation:** 2
**Contribution:** 2
**Rating:** 2
**Confidence:** 4

**Summary:**

This paper proposes using topological data analysis tools (Mapper algorithm and topological deep learning) to improve graph neural network-based multi-omics models for cancer survival prediction. The authors introduce two novel architectures: M-SAN (Mapper with Simplicial Attention Network) and M-HGAT (Mapper with Hypergraph Attention Network). The key claim is that these topology-based approaches outperform both baseline correlation networks and models augmented with biological prior knowledge (protein-protein interactions, gene pathways) across six cancer types. The paper demonstrates that their models learn biologically meaningful gene interactions despite having no access to prior biological knowledge.

**Strengths:**

1. **Comprehensive evaluation**: Testing across 6 cancer types with multiple architectures shows thoroughness.

2. **Practical approach**: Avoiding dependence on curated biological networks is pragmatically valuable given their incompleteness and tissue-specificity issues.

3. **Reproducibility effort**: Code availability promise and detailed hyperparameters support reproducibility.

4. **Interesting negative results**: The analysis of when Mapper helps (Section 4.2, Tables 9-10) provides useful insights about method applicability.

5. **Clear motivation**: The paper articulates the limitations of current approaches well.

**Weaknesses:**

1. **Statistical validation**:
   - Only 5 runs per experiment is insufficient
   - No significance testing between methods
   - Standard errors often overlap between methods
   - No correction for multiple comparisons across 6 cancer types

2. **Experimental design flaws**:
   - Different batch sizes between patient similarity (no batching) and biological interaction models (batched) confounds architectural comparisons
   - Vastly different parameter counts (Table 8: 5M to 414M) make comparison unfair
   - Different training times and convergence criteria may favor different models

3. **Baseline issues**:
   - GGNN comparison lacks experimental parity
   - No comparison with other recent multi-omics methods (only cite Leng et al. 2022 comparison but don't include those methods)
   - Missing comparison with simple baselines (e.g., concatenated features + MLP)

4. **Method clarity**:
   - Novel SAN message passing scheme needs clearer mathematical formulation
   - "Gradient of feature matrix" initialization is undefined
   - Predictive Mapper details missing
   - How are outliers in test set handled exactly?

5. **Biological validation concerns**:
   - Single simplex/hyperedge analyzed per cancer (no robustness check)
   - GO enrichment with Bonferroni correction on single test is underpowered
   - Generic cancer processes found, not novel biology
   - No validation that learned interactions are more meaningful than random feature sets of same size

6. **Overfitting risks**:
   - Small sample sizes (149-333)
   - High-dimensional input (up to 18,790 features)
   - Extensive hyperparameter search
   - Small validation sets for hyperparameter selection
   - Many degrees of freedom (network construction, architecture choices)

7. **Generalization claims unsupported**:
   - Only cancer survival prediction tested
   - All data from single source (TCGA)
   - Claims about other domains (line 60) are speculative

**Questions:**

1. What are the actual p-values from paired statistical tests comparing your methods to baselines within each cancer type?

2. Can you provide ablation studies separating Mapper contribution from TDL contribution?

3. Why do M-SAN and M-HGAT sometimes perform worse than M-GNN (e.g., Melanoma, Lung in Table 2)? What does this tell us about when TDL helps?

4. How sensitive are results to:
   - Number of PCA components in filter function?
   - DBSCAN parameters?
   - Distance correlation threshold selection?
   - Number of Mapper intervals?

5. For biological validation:
   - What GO enrichment do you get for randomly selected feature sets of the same size?
   - Are the identified simplices/hyperedges consistent across the 5 runs?
   - How many simplices/hyperedges show significant enrichment, not just the top one?

6. Can you clarify the "gradient of feature matrix" initialization mathematically?

7. Why are parameter counts so different across models (Table 8)? Shouldn't fair comparison use similar model capacities?

8. Have you tested on any non-cancer datasets or survival prediction tasks outside TCGA?

9. What is the actual implementation of predictive Mapper for test set patients, especially outlier handling?

10. Can you provide learning curves showing validation performance during hyperparameter tuning to assess overfitting?

---

> ### Author Response · Authors · 2025-11-20
>
> We thank the reviewer for detailed comments towards improving the paper, and want to highlight the following changes we have made. We hope these address the reviewer's comments and will be reflected in the final score
>
> **Statistical validation**: We have evaluated the c-index of the Mapper/TDL models against the benchmark GNNs using t-tests. The top models for breast cancer and kidney cancer performed better than both benchmark GNNs (Bonferroni-adjusted p-values < 0.05, #hypotheses=6). The top models for lung, colon and ovarian cancer (and other Mapper/TDL models for breast, lung, kidney and ovarian cancer) performed better than their corresponding benchmark GNN (i.e. we compared patient similarity M-GNN, M-SAN and M-HGAT against the patient similarity B-GNN, etc). The results are updated in Tab 2.
>
> **Mapper vs TDL contribution**: The Mapper-GNN (M-GNN) models use the Mapper algorithm without TDL, and can be considered an ablation study. The analysis in Section 4.2 and Table 10 addresses why M-SAN and M-HGAT sometimes perform worse than M-GNN.
>
> **Baselines**: The two benchmark GNNs using the patient similarity network and biological interaction network were implementations of benchmarks from Leng et al., 2022 [1] based on Wang et al., 2021 [2] and Xing et al., 2021 [3]. We have added results from CustOmics — a VAE based model (c-indices 0.63–0.64), and BulkRNABert — an LLM-based model (c-indices: 0.60--0.65) to Tab 2. These are outperformed by our models (c-indices: 0.67-0.68). We also implemented a simple MLP baseline, with performance added to Tab 2, the c-indices (0.45 - 0.51) are worse than other models.
>
> **Biological validation**: We added Tab 11 in the Appendix, with top significant GO biological processes for a randomly selected subset of nodes. For several cancers the features showed no significant enrichment in any GO biological process, and there were no processes that are clearly linked to cancer progression, or specific cancer type, in contrast to the top features identified by our model.
>
> **Mathematical methods**: We apologize for the missing details and have added them to the methods section. The gradient for rank-k simplices is defined as $B^T_{k+1}X_k$, where  $X_k$ is the feature matrix and $B_{k+1}$ is the incident matrix that indicates which rank-(k+1) simplices neighbor which rank-k simplices. The predictive Mapper algorithm retains the covers/cluster boundaries of the preimage, allowing it to assign new points to an existing node on the Mapper graph, or attribute it as an outlier (does not cluster into an existing node). With an outlier test sample, we append it to the (training set) Mapper graph as a new node, and add edges based on the distance correlation between the test sample’s features and the features in the existing Mapper nodes.
>
> **Overfitting/sensitivity**: We constrained hyperparameters that were not cancer-network specific to be the same across all cancer types. However, due to the differences in connectivity of each cancer network, network-dependent hyperparameters are optimized separately. We added Fig. 4-9 in the Appendix showing hyperparameter importance. In many plots, runtime is more important than other hyperparameters, suggesting that the model is not oversensitive. We included Fig 10 showing validation loss curves for each of the top models.
>
> **Mapper parameters**: To reduce the number of hyperparameters, we used the default Mapper algorithm hyperparameters for PCA and DBSCAN. The number of Mapper intervals determined graph compression and was tuned as a hyperparameter for biological interaction models (see above), and for patient similarity models, higher values were used to prevent over-compression of the patient similarity network. See the above point for distance correlation threshold parameters.
>
> **Experimental design**.
> _Batch size_: the patient similarity and biological interaction approaches are necessarily architecturally different, as the former is a node-prediction task, and the latter is a graph-prediction task. The two approaches are common in the field, and we see value in comparing them on the same dataset, which has not been done.
>
> _Parameter counts_: In order to compare architectures, we kept the same number of layers (GATs/SANs/HGATs) within our model, resulting in different numbers of parameters. However, the best model on average was the biological interaction M-HGAT, which has fewer than 300 parameters, therefore the improvement is not due to parameter count.
>
> [1] D Leng et al. A benchmark study of deep learning-based multi- omics data fusion methods for cancer. Genome biology, 23(1):171, 2022.
>
> [2] T Wang et al. Mogonet integrates multi-omics data using graph convolutional networks allowing patient classi-
> fication and biomarker identification. Nature Comm, 12(1):3445, 2021.
>
> [3] X Xing et. al. An interpretable multi-level enhanced graph attention network for disease diagnosis with
> gene expression data. IEEE BIBM, pp. 556–561. 2021.

---

### Official Review · Reviewer_WFpt · 2025-10-29

**Soundness:** 3
**Presentation:** 3
**Contribution:** 2
**Rating:** 4
**Confidence:** 3

**Summary:**

In this study, the authors proposed to make use Mapper algorithms and topological deep learning (TDL) to improve the sample classification/prediction using multi-omics datasets. The evaluation results showed the improved performance on cancer patient survival dataset.

**Strengths:**

Make use Mapper algorithms and topological deep learning (TDL) to improve the sample classification/prediction using multi-omics datasets.
The evaluation results showed improved performance on cancer patient survival dataset.

**Weaknesses:**

In multi-omics data analysis, the prediction of cancer patients' survival data is not much meaningful (also need to consider the confoundation factors, like age, gender, stage of individual patients).
Rather, the discovery of novel multi-omic features/biomarkers that can explain the mechanisms of patients' survial or drug response are more important. However, the evaluation of the important targets and mechanisms (espeically the multi-omic interactions are not well presented).

**Questions:**

Put more efforts on the identification of multi-omic signaling interactions that are correlated or associated with patients' survival and drug responses are important.
Also considering the confounding factors in the prediction model in addition to multi-omics data.

---

> ### Author Response · Authors · 2025-11-20
>
> We thank the reviewer for the comments, however, the premise of using ML for cancer survival prediction using multiomics data is an expectation that a good model will learn the correct representation of the disease biology to be able to pick up on the relevant signatures of health, aging, drug resistance present in the sequencing data. And while these should be consistent with what is currently known in the field, using ML models that do not require input of biological prior knowledge allows them to potentially uncover new biomarkers and signatures not yet known.
>
> We expect the model to perform even better with additional features such as age, and cancer stage, however evaluating model performance using TCGA data without these attributes is extremely common in the field, see e.g. [1-5], and we respectfully disagree that this is a weakness in our work. We hope this addresses the reviewer's comments and can be reflected in the final score.
>
> [1] Hakim Benkirane, Yoann Pradat, Stefan Michiels, and Paul-Henry Courn`ede. Customics: A versa- tile deep-learning based strategy for multi-omics integration. PLOS Computational Biology, 19 (3):e1010921, 2023.
>
> [2] Maxence G´elard, Guillaume Richard, Thomas Pierrot, and Paul-Henry Courn`ede. Bulkrnabert:
> Cancer prognosis from bulk rna-seq based language models. In Machine Learning for Health
> (ML4H), pp. 384–400. PMLR, 2025.
>
> [3] Dongjin Leng, Linyi Zheng, Yuqi Wen, Yunhao Zhang, Lianlian Wu, Jing Wang, Meihong Wang,
> Zhongnan Zhang, Song He, and Xiaochen Bo. A benchmark study of deep learning-based multi-
> omics data fusion methods for cancer. Genome biology, 23(1):171, 2022.
>
> [4] Chia Yan Tan, Huey Fang Ong, Chern Hong Lim, Mei Sze Tan, Ean Hin Ooi, and KokSheik Wong. Amogel: a multi-omics classification framework using associative graph neural networks with prior knowledge for biomarker identification. BMC bioinformatics, 26(1):1–27, 2025.
>
> [5] Jiening Zhu, Jung Hun Oh, Anish K Simhal, Rena Elkin, Larry Norton, Joseph O Deasy, and Allen
> Tannenbaum. Geometric graph neural networks on multi-omics data to predict cancer survival
> outcomes. Computers in biology and medicine, 163:107117, 2023.

---

### Official Review · Reviewer_Qjjy · 2025-10-30

**Soundness:** 2
**Presentation:** 2
**Contribution:** 2
**Rating:** 2
**Confidence:** 3

**Summary:**

This manuscript addresses the task of multi-omics modeling, specifically focusing on leveraging GNN-based approaches to capture correlations in networks derived from multiple omics layers. Unlike prior methods that incorporate external knowledge such as protein-protein interaction networks or gene pathway information, this work aims to enhance modeling directly from the topological structure of the data. The proposed models, M-SAN and M-HGAT, demonstrate improved performance over selected baselines on the cancer survival prediction task.

**Strengths:**

- The manuscript is well-written and well-argued. It clearly explains the limitations of existing multi-omics GNN models and proposes the motivation to use topological tools to improve network representations.

- This paper introduces a framework that does not require prior knowledge and utilizes the Mapper algorithm and topological deep learning (including a novel simplex attention network) to capture high-order interactions and reduce oversmoothing in related networks. Technically speaking, this makes sense.

- This paper evaluates the M-SAN and M-HGAT models in survival prediction for six cancer types, demonstrating superior performance compared to state-of-the-art GNN models and knowledge augmentation methods. Furthermore, the analysis reveals biologically significant feature interactions.

**Weaknesses:**

- This paper has limited innovation because it focuses on specific tasks in the multi-omics field, with the main goal of improving graph neural network (GNN) models using the Mapper algorithm and topological deep learning. While technically interesting, this approach is incremental and may face scalability challenges on large-scale datasets, and its applicability in other domains or graph-based tasks has not been fully demonstrated.

- The experimental setup is not fully convincing, as only GGNN (Zhu et al., 2023) is used as a baseline. Including additional baselines, particularly non-GNN methods or other multi-omics survival models, would provide a more comprehensive evaluation and strengthen the claims of improvement.

**Questions:**

Please see the weaknesses section for further discussion.

---

> ### Author Response · Authors · 2025-11-20
>
> We thank the reviewer for their comments, and offer the following points of clarification and rebuttal.
>
> **Limited innovation**: We believe that inferring disease outcomes from sequencing data is an important step towards comprehension of biology by AI models. While we built on and made comparisons with previous state of the art models which happened to be GNNs, the problem of understanding sequencing data is much broader and has been attempted using many types of modern architectures, without more success (see point below). To the concern of scalability, models that can extract insights with small datasets are more valuable in biology, where experiments to collect data are more expensive than other areas of ML (for context, the TCGA dataset we use is one of the largest collections of multi-omics data, and is the result of a 12-year joint program between the National Cancer Institute and National Human Genome Research Institute). Furthermore, while we restricted ourself to one application, the patient similarity models carry out node-prediction, while the biological similarity models carry out graph-level prediction, showing that our model is applicable and useful to different GNN types.
>
> **Baselines**: We wanted to highlight that in addition to GGNN, the two benchmark GNNs using the patient similarity network and biological interaction network were implementations of benchmarks from Leng et al., 2022 [1] based on Wang et al., 2021 [2] and Xing et al., 2021 [3]. Additionally, we have added available results from CustOmics — a VAE based model (c-indices 0.63-0.64), and BulkRNABert — an LLM-based model (c-indices: 0.60-0.65) to Table 2, and observe that these are outperformed by our models (c-indices: 0.67 – 0.68).
>
> We hope these address some of the reviewer's concerns and can be reflected in the final score.
>
> [1] Dongjin Leng, Linyi Zheng, Yuqi Wen, Yunhao Zhang, Lianlian Wu, Jing Wang, Meihong Wang, Zhongnan Zhang, Song He, and Xiaochen Bo. A benchmark study of deep learning-based multi- omics data fusion methods for cancer. Genome biology, 23(1):171, 2022.
>
> [2] Tongxin Wang, Wei Shao, Zhi Huang, Haixu Tang, Jie Zhang, Zhengming Ding, and Kun Huang.
> Mogonet integrates multi-omics data using graph convolutional networks allowing patient classi-
> fication and biomarker identification. Nature communications, 12(1):3445, 2021.
>
> [3] Xiaohan Xing, Fan Yang, Hang Li, Jun Zhang, Yu Zhao, Mingxuan Gao, Junzhou Huang, and Jian-
> hua Yao. An interpretable multi-level enhanced graph attention network for disease diagnosis with
> gene expression data. In 2021 IEEE International Conference on Bioinformatics and Biomedicine
> (BIBM), pp. 556–561. IEEE, 2021.

---

### Official Review · Reviewer_EwGi · 2025-10-31

**Soundness:** 3
**Presentation:** 3
**Contribution:** 2
**Rating:** 6
**Confidence:** 3

**Summary:**

This paper proposes two novel GNN-based architectures, Mapper SAN (M-SAN) and Mapper HyperGAT (M-HGAT), that enhance cancer survival prediction from multi-omics data by leveraging topological tools rather than biological priors. The authors apply the Mapper algorithm for unsupervised node pooling to address oversmoothing in correlation networks, and introduce topological deep learning architectures (simplicial attention networks and hypergraph attention networks) to capture higher-order interactions. In the evaluation presented, the proposed methods consistently outperform both naive correlation network approaches and models augmented with protein-protein interaction or gene pathway information.

**Strengths:**

1. The application of topological data analysis (Mapper algorithm) and topological deep learning to address fundamental issues in GNN-based multi-omics modeling is creative and well-justified. The paper clearly articulates the problems of oversmoothing due to dense connectivity and the limitation of pairwise interactions, and proposes targeted solutions that directly address these issues rather than applying generic improvements.
2. The proposed methods outperform GGNN (which leverages explicit biological knowledge including protein-protein interactions and pathway data) across multiple cancer types and metrics. The investigation of when Mapper helps versus when it doesn't (distinguishing "Mapper group" vs. "Benchmark group" cancers) and the mechanistic analysis through GNNExplainer application demonstrates scientific rigor.
3. The post-hoc analysis using GNNExplainer to validate learned interactions is helpful.

**Weaknesses:**

1. With sample sizes ranging from 149-333 and modest performance improvements in several cases, the statistical significance of reported gains is questionable. Although error bars are provided, no formal significance testing is reported.
2. The Mapper algorithm and TDL models introduce numerous hyperparameters (filter intervals: 5-20, overlap fractions: 0.1-0.0001, distance correlation thresholds, max_rank values), each tuned separately per cancer type. Table 7 shows dramatic variation across cancers (e.g., mRNA intervals from 0.1 to 6.0), suggesting either the method is highly sensitive to these choices or the tuning process introduces implicit cancer-specific adaptation that could inflate performance estimates. The paper lacks ablation studies on sensitivity to these hyperparameters or justification for the tuning strategy. This could lead to overfitting to the specific cancer types in the dataset.

**Questions:**

1. Given the small sample sizes and multiple comparisons across six cancer types, could you provide formally tested statistical significance (with appropriate correction for multiple testing) for the key performance differences?
2. The initialization of higher-rank features as "gradients" and the behavior of the predictive Mapper algorithm for outliers need clarification. How exactly are these implemented? Does outlier handling in the test set affect performance?
3. How do your methods compare to simpler alternatives like dimensionality reduction followed by standard classifiers?
4. Can you provide negative control analyses (e.g., enrichment p-values for randomly selected gene sets from your learned features)? Which identified interactions are truly novel versus confirmatory of known biology? Have you validated any predicted interactions experimentally or against curated databases? (searching in the databases will be a quick thing, and provide a strong evidence)

---

> ### Author Response · Authors · 2025-11-20
>
> We thank the reviewer for the positive appraisal of our work, and helpful comments towards improving it. We wish to highlight some of the changes we have incorporated to address the reviewer’s concerns and hope these can be reflected in the final score.
>
> **Significance testing**: We have evaluated the c-index of the Mapper/TDL models against the benchmark GNNs using t-tests. The top models for breast cancer and kidney cancer performed better than both benchmark GNNs (Bonferroni-adjusted p-values < 0.05, #hypotheses=6). The top models for lung cancer, colon cancer and ovarian cancer (and several other Mapper/TDL models for breast, lung, kidney and ovarian cancer) performed better than their corresponding benchmark GNN (i.e. we tested the performance of the patient similarity M-GNN, M-SAN and M-HGAT against the patient similarity B-GNN, and the same for the biological interaction models). The results are updated in Tab 2.
>
> **Gradients and predictive Mapper**: We apologize for the missing details and have added them to the methods section. In brief, a for rank-k simplifies, we have the feature matrix $X_k$. The gradient is defined as $B^T_{k+1}X_k$, where $B_{k+1}$ is the incident matrix that indicates which rank-(k+1) simplices are in the neighborhood of which rank-k simplices [1]. The predictive Mapper algorithm retains the edges of the covers and cluster boundaries of the preimage, which allows it to assign new points to an existing node on the computed Mapper graph, or attribute it as an outlier (doesn't cluster into an existing node), allowing us to apply our model to the inductive case. With an outlier test sample, we append it to the (training set) Mapper graph as a new node, and add edges based on the correlation between the test sample’s features and the features in the existing Mapper nodes. In the patient similarity Mapper graph, there are frequently nodes consisting of single samples, so this treatment of test samples is consistent with the treatment of training samples.
>
> **Simple baseline comparison**: We have implemented a simple MLP baseline consisting of a linear layer + activation (LeakyRelu) for dimensionality reduction of each of the omics types, followed by concatenation and a final MLP layer for classification. The performance of the MLP baseline is added to Table 2, and the c-indices (0.45 – 0.51) are worse than the other models. In addition, we have included the performance of CustOmics, a VAE-based multi-omics model (c-indices 0.63-0.64), and BulkRNABert, an LLM-based single-omics model (c-indices: 0.60-0.65) on the breast cancer and lung cancer datasets. Both of these models are also outperformed by our models (c-indices: 0.67 – 0.68).
>
> **Negative controls**: We have added Tab 11 in the Appendix, with the top significant (Bonferroni-adjusted p-values < 0.05) GO biological processes for a randomly selected subset of nodes from the same model and omics network and corresponding to a similar number of gene features for each cancer type as in Tab 4. The features for breast and lung cancer did not show significant enrichment in any GO biological process, and ovarian cancer showed significant enrichment in only one GO biological process. For the other cancers, other than general processes related to transcription and translation, there are no processes that are clearly linked to cancer progression, or to the specific cancer type, in contrast to the top features identified by our model.
>
> **Hyperparameter tuning and overfitting**: When tuning hyperparameters, we constrained hyperparameters that were not cancer-network specific (dropout, learning rate, weight decay, number of epochs) to be the same across all cancer types. However, due to the differences in connectivity of each cancer omics network, we optimized the network-dependent hyperparameters for each cancer type separately. Hyperparameter optimization was done using a grid search using the Weights & Biases tool, and we have added Fig. 4-9 in the Appendix showing the importance and correlations of the hyperparameters. In many of the plots, it can be seen that runtime is more important than several of the hyperparameters, suggesting that the model is not oversensitive to hyperparameters. Additionally, we have included Figure 10 showing validation loss curves during training for each of the top models, to address the concern of overfitting. Finally, the mRNA intervals are chosen to yield a median of 1-10 neighbors for each node, and this depends on how much the network is compressed by Mapper (by the #intervals parameter), therefore while the values for mRNA/DNA Methy/miRNA seem widely different, the degree distributions are not that dissimilar. We have added these points to the Appendix.
>
> [1] Claudio Battiloro, Lucia Testa, Lorenzo Giusti, Stefania Sardellitti, Paolo Di Lorenzo, and Sergio Barbarossa. Generalized simplicial attention neural networks. IEEE Transactions on Signal and Information Processing over Networks, 2024.

---

### Meta-Review · Area_Chair_meDz · 2026-01-06

**Summary:**

Reviewers have raised concerns about the limited methodological innovation, concerns about statistical significance of the evaluation outcomes, insufficient comparison against existing methods baselines, limited evaluations based on small samples, and the large number of hyperparameters that need tuning and potential sensitivity of the outcomes to the choice of these parameters.
In response, the authors have provided their responses to the reviewers' concerns, including the statistical significance tests, hyperparameter tuning, provision of additional comparisons against more baselines (including a simple ML model proposed by one of the reviewers).
Although the authors rebuttal seems to address several concerns raised by the reviewers, major concerns about the lack of significant methodological innovations appears to remain despite the authors rebuttal. Concerns about limited biological validation and novel biological insights do not seem to have been sufficiently addressed.

**Reviewer Concerns:**

I believe concerns about hyperparameter tuning and the statistical significance of the results have been addressed reasonably.
However, concerns regarding limited methodological innovation/contribution and insufficient biological validation and novel biological insights are still outstanding.

**Reviewer Scores:**

Reviewer EwGi gave the highest score among the reviewers, and I think it is likely that the reviewer may have maintained the score, which is already higher than those from all other reviewers.

Reviewer Qjjy' concerns haven't been addressed in a sufficient manner and major concerns appear to be still outstanding.
The additional benchmarks may have slightly increased the score from 2 to 3 or so.

The rebuttal addressed Reviewer WFpt's concerns only indirectly and it's likely that the reviewer may have mantained the original score.

Finally, 8VSi had substantial concerns and questions about the manuscript, which the authors attempted to address to some extent.
This may have increased the score from 2 to 3 or so, but major concerns are likely to remain.

---

### Decision · Program_Chairs · 2026-01-26

Reject